# Mapping CircRNA–miRNA–mRNA regulatory axis identifies hsa_circ_0080942 and hsa_circ_0080135 as a potential theranostic agents for SARS-CoV-2 infection

**Hassan Ayaz[1]⊙, Nouman Aslam[1]⊙, Faryal Mehwish Awan[1]\*, Rabea Basri[1], Bisma Rauff[2], Badr Alzahrani[3], Muhammad Arif[1], Aqsa Ikram[4], Ayesha Obaid[1], Anam Naz[4], Sadiq Noor Khan[1], Burton B. Yang[5,6,7], Azhar Nazir[1]**

**1** Department of Medical Lab Technology, The University of Haripur (UOH), Haripur, Khyber Pakhtunkhwa, Pakistan, **2** Department of Biomedical Engineering, University of Engineering and Technology (UET), Lahore, Narowal, Pakistan, **3** Department of Clinical Laboratory Sciences, College of Applied Medical Sciences, Jouf University, Sakaka, Saudi Arabia, **4** Institute of Molecular Biology and Biotechnology (IMBB), The University of Lahore (UOL), Lahore, Pakistan, **5** Sunnybrook Research Institute, Sunnybrook Health Sciences Centre, Toronto, Canada, **6** Department of Laboratory Medicine and Pathobiology, University of Toronto, Toronto, Canada, **7** Institute of Medical Sciences, University of Toronto, Toronto, Canada

⊙ These authors contributed equally to this work.
\* faryal.mehwish@uoh.edu.pk, faryal_mehwish@yahoo.com

**Data Availability Statement:** Availability of Data and Materials The authors declare that the data supporting the findings of this study are addressed

## Abstract

Non-coding RNAs (ncRNAs) can control the flux of genetic information; affect RNA stability and play crucial roles in mediating epigenetic modifications. A number of studies have highlighted the potential roles of both virus-encoded and host-encoded ncRNAs in viral infections, transmission and therapeutics. However, the role of an emerging type of non-coding transcript, circular RNA (circRNA) in severe acute respiratory syndrome coronavirus 2 (SARS-CoV-2) infection has not been fully elucidated so far. Moreover, the potential pathogenic role of circRNA-miRNA-mRNA regulatory axis has not been fully explored as yet. The current study aimed to holistically map the regulatory networks driven by SARS-CoV-2 related circRNAs, miRNAs and mRNAs to uncover plausible interactions and interplay amongst them in order to explore possible therapeutic options in SARS-CoV-2 infection. Patient datasets were analyzed systematically in a unified approach to explore circRNA, miRNA, and mRNA expression profiles. CircRNA-miRNA-mRNA network was constructed based on cytokine storm related circRNAs forming a total of 165 circRNA-miRNA-mRNA pairs. This study implies the potential regulatory role of the obtained circRNA-miRNA-mRNA network and proposes that two differentially expressed circRNAs hsa_circ_0080942 and hsa_circ_0080135 might serve as a potential theranostic agents for SARS-CoV-2 infection. Collectively, the results shed light on the functional role of circRNAs as ceRNAs to sponge miRNA and regulate mRNA expression during SARS-CoV-2 infection.

within the article and confirm that all the datasets analyzed during the current study are accessible from the literature as well as from the GEO database (https://www.ncbi.nlm.nih.gov/gds/) with accession details (GSE166552, GSE19137, PRJCA002617, (McDonald, Enguita et al. 2021), (Arora, Singh et al. 2020), (Farr, Rootes et al. 2021), (Li, Hu et al. 2020), (Chow and Salmena 2020), (Demirci and Demirci 2021), (Chi, Ge et al. 2020), (Lin, Luo et al. 2020), (Chen, Wu et al. 2020), (Chen, Liu et al. 2020), (Blanco-Melo, Nilsson-Payant et al. 2020), (Li, Hu et al. 2020), (Del Valle, Kim-Schulze et al. 2020), (Qin, Zhou et al. 2020), (Yang, Shen et al. 2020), (Dhar, Vishnupriyan et al. 2021), and (Huang, Wang et al. 2020) datasets).

**Funding:** The authors extend their appreciation to the Deputyship for Research & Innovation, Ministry of Education in Saudi Arabia for funding this research work through the project number 223202.

**Competing interests:** The authors have declared that no competing interests exist.

## Introduction

Severe acute respiratory syndrome coronavirus 2 (SARS-CoV-2), an enveloped RNA virus with a genome size of 29,903 bp, is a highly infectious and pathogenic coronavirus [1]. Initially, on December 31, 2019, China reported unusual viral pneumonia outbreak in the city of Wuhan. Since then, the virus SARS-CoV-2 that causes the COrona VIrus Disease 2019 (COVID-19) has evolved into a pandemic. Currently, >750 million people have been infected globally while >6.8 million people have lost their lives due to COVID-19 (World Health Organization, February 22, 2023). Once inside the host cell, SARS-CoV-2 triggers strong innate immune response with excessive production and infiltration of pro-inflammatory cytokines resulting in COVID-19 associated cytokine storm [2]. A cytokine storm is an umbrella term which describes a kind of hyper-inflammatory reaction that can result in life threatening systemic inflammation, immune dysregulation and multi-organ failure if left untreated [3]. These inflammatory mediators are being intricately controlled by regulatory machinery employed by the host cell [4]. Researchers initially focused upon studying disease pathogenesis and development of vaccines in response to COVID-19 pandemic. However, currently more focus has shifted towards elucidating the molecular mechanisms involved in various pathological responses exhibited during the course of infection. It is postulated that differentially expressed non-coding RNAs (ncRNAs) might be implicated in the regulation of the cellular processes which regulate the SARS-CoV-2 pathogenicity and cytokine-mediated immune responses [5]. Current study focuses on the premise that the variations in expression of these ncRNAs may possibly present a novel avenue to explore the pathogenesis of COVID-19.

Recent advances in high-throughput sequencing technologies and computational methods have discovered a substantial number of ncRNA ultimately providing new insights into their role in a range of human diseases [6, 7]. Various studies have shown that the ncRNAs, which include microRNAs (miRNAs), long non-coding RNAs (lncRNAs), and circular RNAs (circRNAs) play a crucial role in the progression of viral diseases [7–10]. miRNAs, one of the type of ncRNA are quite well studied and are known to play a pivotal role in the regulation of many genes, especially those at intersections of signaling pathways involved in development and growth control [11, 12]. Interestingly, viruses have been reported to interact with cellular miRNAs to manipulate both viral and cellular gene expression as well as to augment their replication potential [7, 13]. Moreover, miRNAs regulate viral persistence, host immune evasion and long-term survival in the host cell [14, 15]. Likewise, the expression of cellular lncRNAs may also be altered in response to viral replication inside the cell. It has been reported that many cellular lncRNAs are expressed in response to the antiviral pathways activated by viral infections [10, 16]. It is important to note that lncRNAs utilized by the viral machinery to enhance its replication could be those lncRNAs which are up-regulated in response to viral infections. Similarly, to counteract viral infection, the host cell has evolved to generate various cellular lncRNAs [10].

On the other hand, circRNAs, a novel class of ncRNAs have been reported to play crucial role in regulating viral infections and their dysregulation has been implicated in the pathogenesis of various diseases [17, 18]. While the biological functions of the majority of circRNAs are still not established, accumulating piece of evidence confirms that circRNAs perform many regulatory functions via translocating or sequestering proteins, exerting transcriptional and translational control as well as facilitating interactions between proteins [18]. circRNAs play multifunctional roles hence they are implicated in a range of biological and pathological processes that may affect the progression of several diseases including viral infections [19, 20]. Moreover, circRNAs modulate the innate immune responses during viral infections as well. Additionally, circRNAs have also been recommended as biomarkers for differentiating viral from non-viral infections [21, 22].

The circRNA–miRNA–mRNA regulatory axis has been shown to be of high importance in association with several human diseases including cancers, diabetes, Alzheimer's disease, and cardiovascular diseases [23–25]. These networks are involved in signaling pathways of different human diseases by regulating expression profiling of pathogenicity-related genes [24]. In essence, circRNAs regulates the expression of downstream target genes of miRNA, in turn miRNA can decrease the stability of the mRNA or in other words inhibit the translation of the mRNA by targeting its 3′-untranslated region (UTR), thus, negatively regulating the expression of the target genes [26]. Increasing evidence has suggested that this regulatory axis might also be involved in regulating crucial mechanisms in immune related response to SARS-CoV-2 [27].

There is an abundance of COVID-19 related transcriptomics studies and data, however, their use is limited by the confounding factors pertaining to each study. In the current study, we have analyzed different datasets in a unified approach which might help in understanding the molecular basis of COVID-19. Moreover, reverse engineering approach was utilized to derive regulatory interactions between circRNAs, miRNAs and mRNAs from gene expression data of SARS-CoV-2 patients. In order to gain better understanding of molecular and immuno-pathological basis, possible regulatory mechanisms of circRNA-miRNA-mRNA axis during SARS-CoV-2 infection were investigated. The circRNA–miRNA–mRNA regulatory network consisting of differentially expressed circRNAs and their downstream miRNAs and target mRNAs have been constructed for SARS CoV-2 related pathogenesis. The circRNAs that may play critical roles in regulating the cytokine storm during SARS-CoV-2 infection were identified. The results from this study revealed some candidate circRNAs that might function as potential theranostic agents in SARS-CoV-2 infection. Moreover, targeting the "cytokine storm" using circRNAs might be a feasible therapeutic approach to combat COVID-19 (Fig 1). Together, this research provides new insights into the triple regulatory network controlled by the circRNA–miRNA–mRNA regulatory axis. The results also provide possible understandings on the roles of circRNAs in host-virus interactions and will facilitate research studies on SARS-CoV-2 infection and pathogenesis in the future.

## Materials and methods

### Data acquisition and processing

The Gene Expression Omnibus (GEO) [28] (https://www.ncbi.nlm.nih.gov/geo/), a database supported by the National Center for Biotechnology Information (NCBI) at the National Library of Medicine (NLM) was used to access the microarray and RNA-sequencing datasets that contains circRNA, miRNA and mRNA expression profiles of SARS-CoV-2 infected patients at various stages (Table 1) (Fig 2). GEO accepts high-throughput gene expression profiles (both raw and processed data) along with sample characteristics, methodology, and experimental research design [29]. Two circRNA datasets including GSE166552 [17] and PRJCA002617 [30] were retrieved after comprehensive screening. Six miRNA datasets including [27, 31–35] were screened. Moreover, fifteen datasets of mRNA expression profiles were retrieved including GSE19137, GSE166552, PRJCA002617, [32, 33, 36–45] datasets. We also downloaded datasets from national genomic data center (https://ngdc.cncb.ac.cn/). As the present study does not involve human subjects and due to free availability of data in the GEO database, neither ethical approval nor informed consent was required.

The GSE166552 dataset included 06 samples (03 SARS-CoV-2 positive and 03 controls). The PRJCA002617 dataset included 24 samples (12 SARS-CoV-2 positive and 12 controls). McDonald et al., dataset included 45 samples (25 SARS-CoV-2 infected patient samples and 20 healthy controls). The GSE19137 dataset included 21 samples (03 negative and 18 positive

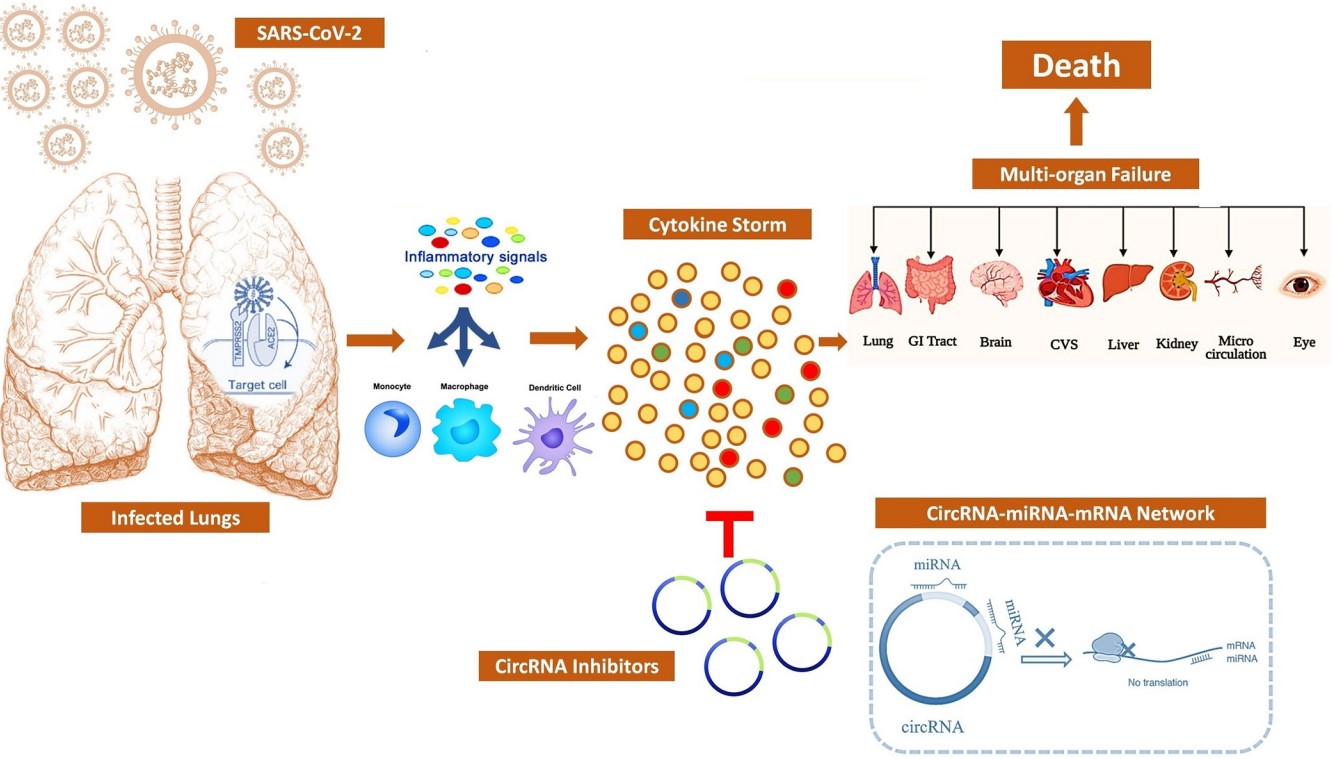

**Fig 1. Using circRNA based therapeutics to mitigate cytokine storm syndrome induced by SARS-CoV-2.**

for SARS-CoV). Chow et al., dataset included 249 samples (147 SARS-CoV-2 infected samples and 102 controls). Dhar et al., dataset included 2157 samples (including 915 severe COVID-19 patients). Liu et al., dataset contained 40 samples in his study (including 13 severe COVID-19 patients). Farr et al., dataset included 20 samples (10 COVID-19 patients and 10 age and gender matched healthy controls). Li et al., dataset included 14 samples (10 COVID-19 patients and 4 healthy donors). Huang et al., dataset comprised of 41 samples (including 13 severe COVID-19 patients). Chi et al., dataset included 70 SARS-CoV-2 infected patients, 04 convalescent cases and 04 healthy controls. Lin et al., dataset included 334 samples in their study (including 23 severe COVID-19 patient samples). Chen et al(b)., dataset included 21 samples (including 11 severe COVID-19 patient samples). Chen et al(c)., dataset study contained 29 samples (including 14 severe COVID-19 patient samples). Blenco Melo et al., included 48 samples (24 SARS-CoV-2 positive samples and 24 negative samples). Del Velle et al., dataset included a total of 1484 samples (1097 positive for SARS-CoV-2 infection and 387 controls). Qin et al., dataset comprised of 452 samples in their study (including 286 severe COVID-19 patient samples). Yang et al., dataset included 50 samples (including 36 severe COVID-19 patient samples).

## Tools for the prediction of circRNA, miRNA and mRNA targets

For the prediction of circRNA targets by using miRNA as an input search, we used different comprehensive databases including CircBank, CircInteractome and RNAInter v4.0 web tools (Table 2). CircBank (http://www.circbank.cn/) is a comprehensive, publicly available, functionally annotated human circRNAs database containing information of about 140,000 circRNAs from many different sources [47]. The Users can access information regarding

**Table 1. Datasets used for retrieving expression profiles of circRNAs, miRNAs and mRNAs.**

| Accession no/ Study details | circRNA/ miRNA/ mRNA | Sample Source | Total dysregulated (n) | Up-regulated | Down-regulated |
|---|---|---|---|---|---|
| GSE166552 | CircRNA | Whole Blood Samples | 70 | 30 | 40 |
| PRJCA002617 | CircRNA | Lung Tissue Samples | 31 | 12 | 19 |
| [31] | miRNA | Blood Samples Nasopharyngeal Samples | 8 | 1 | 7 |
| [27] | miRNA | Lung Tissue Samples | 1 | - | 1 |
| [32] | miRNA | Blood Samples Nasopharyngeal Samples | 112 | 43 | 69 |
| [33] | miRNA | Blood Samples | 6 | 3 | 3 |
| [34] | miRNA | Lung Tissue Samples | 6 | 2 | 4 |
| [35] | miRNA | Lung Tissue Samples | 10 | 5 | 5 |
| GSE166552 | mRNA | Blood Samples | 25 | 11 | 14 |
| GSE19137 | mRNA | Lung Tissue Samples | 171 | 108 | 63 |
| PRJCA002617 | mRNA | Lung Tissue Samples | 30 | 16 | 14 |
| [36] | mRNA | Blood Samples | 11 | 11 | - |
| [37] | mRNA | Blood Samples | 6 | 6 | - |
| [38] | mRNA | Blood Samples | 6 | 6 | - |
| [39] | mRNA | Blood Samples | 2 | 2 | - |
| [40] | mRNA | Lung Tissue Samples | 8 | 8 | - |
| [46] | mRNA | Blood Samples | 4 | 4 | - |
| [41] | mRNA | Blood Samples | 3 | 3 | - |
| [42] | mRNA | Blood Samples Nasopharyngeal Samples | 4 | 4 | - |
| [43] | mRNA | Blood Samples | 14 | 14 | - |
| [32] | mRNA | Blood Samples Nasopharyngeal Samples | 3 | 3 | - |
| [44] | mRNA | Nasopharyngeal Samples | 6 | 6 | - |
| [45] | mRNA | Blood Samples | 8 | 8 | - |

conservation status, miRNA targets as well as protein coding potential of query circRNAs [47]. CircInteractome (https://circinteractome.nia.nih.gov/) is a readily accessible web tool for mapping miRNAs and protein-binding sites on junctions as well as junction-flanking sequences of human circRNAs [48]. RNA Interactome Database, RNAInter v4.0 (http://www.rnainter.org/) is a comprehensive RNA-associated interactome platform containing information of more than 41 million interactions of cellular RNAs in 154 species with evidence from both computational and experimental sources [49].

For the prediction of miRNA/ mRNA targets by using mRNAs/miRNAs as an input search, we used databases including miRDB, miRWalk 2.0, miRTarBase, and TargetScan 7.0. miRDB (http://mirdb.org/), is an integrative, freely accessible, open platform for the prediction of miRNA targets. miRNA-target interactions with scores ≥80.0 were considered relevant, statistically significant and with higher confidence in the interactions whereas miRNA-target interactions with scores ≤80.0 were considered not relevant. By utilizing high-throughput experimental data, miRDB predicts miRNA targets in five species along with integrative analysis of gene ontology (GO) data [50]. miRWalk 2.0 (http://mirwalk.umm.uni-heidelberg.de/) provides information of more than 949 million computationally predicted as well as experimentally validated miRNA-mRNA interactions. In order to ensure reliability and accuracy of forecast results, miRWalk 2.0 incorporates 12 algorithms for prediction including miRWalk, mirbridge, Targetscan, Microt4, PITA, Pictar2, RNAhybrid, RNA22, miRNAMap, miRanda, miRMap and miRDB [51]. Cut-off value with a binding score of > 0.95 was considered as a screening threshold. miRTarBase (https://miRTarBase.cuhk.edu.cn/~miRTarBase/miRTarBase_2022/php/index.php) is a manually curated database containing information of

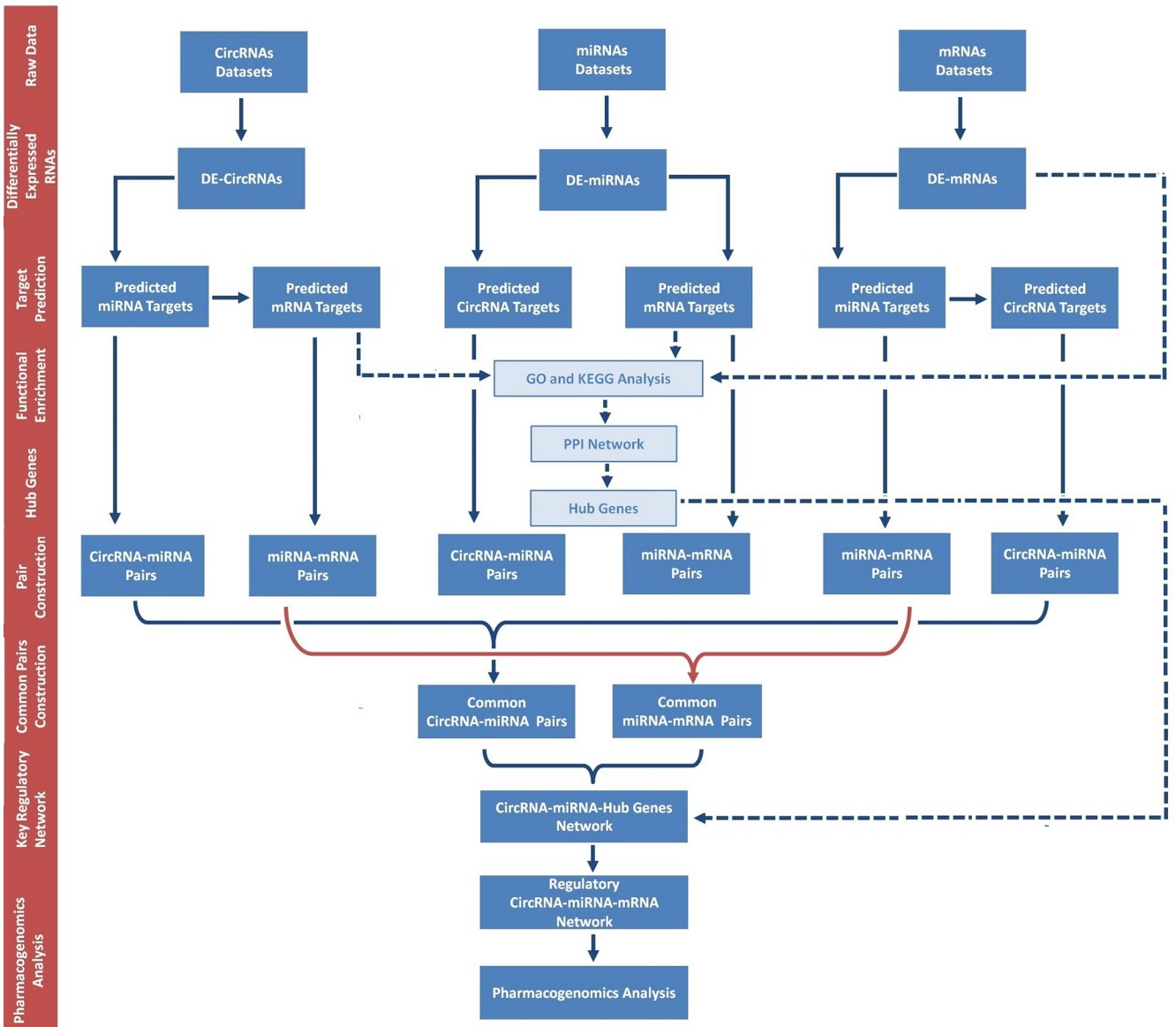

**Fig 2. Flow chart of the approach utilized in the present study for the construction of SARS-CoV-2 related circRNA-miRNA-mRNA regulator network.**

more than 360,000 experimentally validated miRNA-mRNA interactions [52]. miRNA-mRNA interactions have been validated experimentally using microarray, CLIP-seq technology, reporter assays, high through-put sequencing and western blot experiments [52]. All the targets identified via miRTarBase were selected for further analysis. TargetScan v7.0 (http://www.targetscan.org/vert_80/), a flexible web based tool, predicts sequence based effective regulatory targets of miRNAs by incorporating 14 different features [53]. Conservation aggregate score of > 0.80 was considered as selection criteria as this score provides low false discovery rates. An overlap in at least two databases was used as filtering criteria for prioritizing and considering potential candidate targets. Previous comparative studies conducted on miRNA target prediction programs suggested that no program performed consistently superior to all others. Indeed, it has become a common practice for researchers to look at predictions produced by

**Table 2. List of all softwares and tools utilized in the current study.**

| Software | Link | References |
| --- | --- | --- |
| GEO | https://www.ncbi.nlm.nih.gov/gds | [29] |
| GEO2R | https://www.ncbi.nlm.nih.gov/geo/geo2r/ | [54] |
| CircInteractome | https://circinteractome.nia.nih.gov/ | [48] |
| Circbank | http://www.circbank.cn/ | [47] |
| RNAinter v4.0 | http://www.rnainter.org/ | [49] |
| miRTarBase | https://miRTarBase.cuhk.edu.cn/ | [55] |
| miRWalk 2.0 | http://mirwalk.umm.uni-heidelberg.de/ | [51] |
| TargetScan v7.0 | http://www.targetscan.org/vert_80/ | [53] |
| miRDB | http://www.mirdb.org/ | [56] |
| STRING | https://string-db.org/ | [57] |
| Cytoscape | https://cytoscape.org/ | [58, 59] |
| cytoHubba | https://cytoscape.org/ | [60] |
| MCODE | https://cytoscape.org/ | [59] |
| DAVID v6.8 | https://david.ncifcrf.gov/ | [61] |
| KEGG | https://www.genome.jp/kegg/ | [62] |
| PharmGKB | https://www.pharmgkb.org/ | [63] |

different miRNA-target prediction programs and focus on their intersection which might enhances the performance of analyses as well as improve prediction precision. The differences between algorithms are mostly seen in their respective weaknesses, i.e., the subset of false positives. For that reason, the fundamental motivation to focus selectively on the shared prediction by two algorithms is to eliminate false positives while preserving the vast majority of true positive RNAs. Therefore, conclusively, predictions are much more reliable when two or more prediction algorithms are combined, and the minimal loss of true positives are greatly outweighed by the removal of false positives. Selection criteria, threshold and prediction scores for each database were selected on the basis of their previously reported relationship with low false discovery rate and high accuracy in experimental validation studies via PCR and Luciferase assays.

## Data preprocessing and identification of differentially expressed circRNAs, miRNAs, and mRNAs

Background noise correction and quantile normalization of preliminary data were performed. Literature mining and R-base statistical software, GEO2R was utilized for examining the raw gene expression data as well as for the analysis of differential expression profiles of miRNAs, circRNAs and mRNAs. For the datasets retrieved from GEO database, we have utilized GEO2R tool which uses force normalization by applying quantile normalization to the expression data making all selected samples having identical value distribution. For determining whether selected samples were suitable for differential expression analysis, we checked distribution of samples by observing the median-centered values. Median-centered values are generally indicative that the data are normalized and cross-comparable. In summary, data preprocessing was done via log2 transformation, quantile normalization and base line transformation using the median of the samples. The expression fold change was expressed as base-2 logarithm of FC (log2FC) to normalize the expression values obtained from different platforms. For the calculation of false discovery rate and p-value, the GEO2R inbuilt methods were utilized. Differentially expressed genes were considered up-regulated if they met the cut-off criteria of adjusted $P < 0.05$ and $|logFC| >= 1$. For down-regulated ones, $logFC <= -1$

was considered. LogFC means Log2-fold change between two experimental conditions or two groups of Samples. Next, an interactive Venn diagram drawing tool, Venny 2.1.0 (https://bioinfogp.cnb.csic.es/tools/venny/) was used to generate Venn diagram to find the intersection among circRNAs, miRNAs, and mRNAs datasets [64].

Previous studies conducted on analysis of RNAs and their targets suggested that no program performed consistently superior to all others. Indeed, it has become a common practice for researchers to look at predictions produced by prediction programs and focus on common findings which might enhances the performance of analyses as well as improve prediction precision. The algorithms mostly agree on highly expressed RNAs, however, in many cases, algorithm-specific false positives with high read counts are predicted, which is resolved by using the shared output from two (or more) algorithms.

## Pathway and functional enrichment analysis

The Database for Annotation, Visualization and Integrated Discovery database (DAVID version 6.8; https://david.ncifcrf.gov/), a comprehensive knowledge-base functional classification and agglomeration algorithm was used to perform KEGG (Kyoto Encyclopedia of Genes and Genomes) pathway and GO analysis [61]. In order to increase reliability of results, DAVID incorporates 14 annotation categories including BioCarta Pathways, Swiss-Prot Keywords, Molecular Function, cellular components, KEGG Pathways, Biological Process, UniProt Sequence Features, BBID Pathways, SMART Domains, PIR SuperFamily Names, and InterPro Domains along with NIH Genetic Association DB. A p-value less than 0.05 ($\leq 0.05$) was considered to indicate a statistically significant difference [61]. KEGG is a manually curated bioinformatics resource for deciphering high-level cellular and organism-level functions [65].

## Establishment of PPI networks and module analysis

In order to examine and explore the relationship and association among the differentially expressed genes from the retrieved datasets, we constructed a protein-protein interaction (PPI) network by utilizing the STRING webserver (https://string-db.org) [57]. STRING database currently covers more than 9 million proteins from 2,031 organisms. It constructs a PPI network on the basis of direct physical or indirect functional associations. Differentially expressed genes were imported in to STRING database. Afterwards, Cytoscape (https://cytoscape.org/), an open source bioinformatics resource was used to analyze and visualize the molecular interaction networks as well as hub genes via importing the obtained source files from STRING [58]. A confident interaction score $\geq$0.4 was fixed as cutoff standard. Moreover, the cytoHubba plug-in in Cytoscape was used to search the list of hub genes from the PPI network with node degrees [60]. In addition, via using MCODE plug-in, key modules were then screened and assessed from the PPI network [59].

## Construction of circRNA–miRNA–mRNA network

A circRNA–miRNA–mRNA regulatory network was also constructed by using Cytoscape software.

## Statement

All materials and methods were performed in accordance with the relevant guidelines.

## Results

### Differentially expressed circRNAs

GEO2R along with data mining tools were utilized to generate differentially expressed circRNA, miRNA and mRNA profiles. Two SARS-CoV-2 related circRNA datasets including GSE166552 and PRJCA002617 were retrieved after comprehensive screening. By analyzing the data using GEO2R, differentially expressed circRNAs were identified. A total of 70 dysregulated circRNAs were identified from GSE166552 dataset. In addition, a total of 31 dysregulated circRNAs were identified from PRJCA002617 dataset. Comprehensive analysis of two datasets revealed a total of 101 differentially expressed circRNAs (Table 3) (Fig 3).

### Differentially expressed miRNAs

For the prediction of differentially expressed miRNAs, GEO database and data mining tools were utilized. Six different miRNA datasets including McDonald et al., 2021, Arora et al., 2020, Farr et al., 2021, Li et al., 2020a, Chow et al., 2020 and Demirci et al., 2021 were screened. Findings revealed 112 dysregulated miRNAs in Farr et al., 2021 dataset. Dataset of Demirci et al., 2021, revealed 10 dysregulated miRNAs. In another dataset McDonald et al., 2021, 8 dysregulated miRNAs were retrieved. In another dataset of Li et al., 2020, 6 dysregulated miRNAs were retrieved. In addition, Arora et al., 2020 dataset revealed 1 dysregulated miRNA. Furthermore, a dataset of Chow et al., 2020, revealed 6 dysregulated miRNAs. Comprehensive analysis of six different miRNA datasets revealed 143 differentially expressed miRNAs.

**Table 3. List of count of differentially expressed RNAs and their predicted targets.**

| Datasets | Dysregulated circRNAs/miRNAs/mRNAs | | Predicted Targets | | |
| --- | --- | --- | --- | --- | --- |
| | Up-regulated | Down-regulated | circRNAs | miRNAs | mRNA |
| GSE166552 | 30 circRNAs | 40 circRNAs | _ | 4316 | 704000 |
| PRJCA002617 | 12 circRNAs | 19 circRNAs | _ | 1516 | |
| McDonald et al., 2021 | 1 miRNA | 7 miRNAs | 38937 | _ | 77722 |
| Demirci et al., 2021 | 5 miRNAs | 5 miRNAs | | _ | |
| Farr et al., 2021 | 43 miRNAs | 69 miRNAs | | _ | |
| Li et al., 2020 | 3 miRNAs | 3 miRNAs | | _ | |
| Chow et al., 2020 | 2 miRNAs | 4 miRNAs | | _ | |
| Arora et al., 2020 | 1 miRNAs | | | _ | |
| GSE19137 | 108 mRNAs | 63 mRNAs | 858423 | 5109 | _ |
| GSE166552 | 11mRNAs | 14 mRNAs | | | _ |
| PRJCA002617 | 16 mRNAs | 14mRNAs | | | _ |
| Chi et al., 2020 | 11mRNAs | _ | | | _ |
| Chen et al., 2020b | 5 mRNAs | 1 mRNAs | | | _ |
| Lin et al., 2020 | 6 mRNAs | _ | | | _ |
| Chen et al., 2020c | 2 mRNAs | _ | | | _ |
| Blanco-Melo et al., 2020 | 8 mRNAs | _ | | | _ |
| Liu et al., 2020 | 4 mRNAs | _ | | | _ |
| Del Valle et al., 2020 | 3 mRNAs | _ | | | _ |
| Qin et al., 2020 | 4 mRNAs | _ | | | _ |
| Yang et al., 2020 | 14 mRNAs | _ | | | |
| Farr et al., 2021 | 3 mRNAs | _ | | | _ |
| Dhar et al., 2021 | 6 mRNAs | _ | | | _ |
| C. Huang et al., 2020 | 8 mRNAs | _ | | | _ |

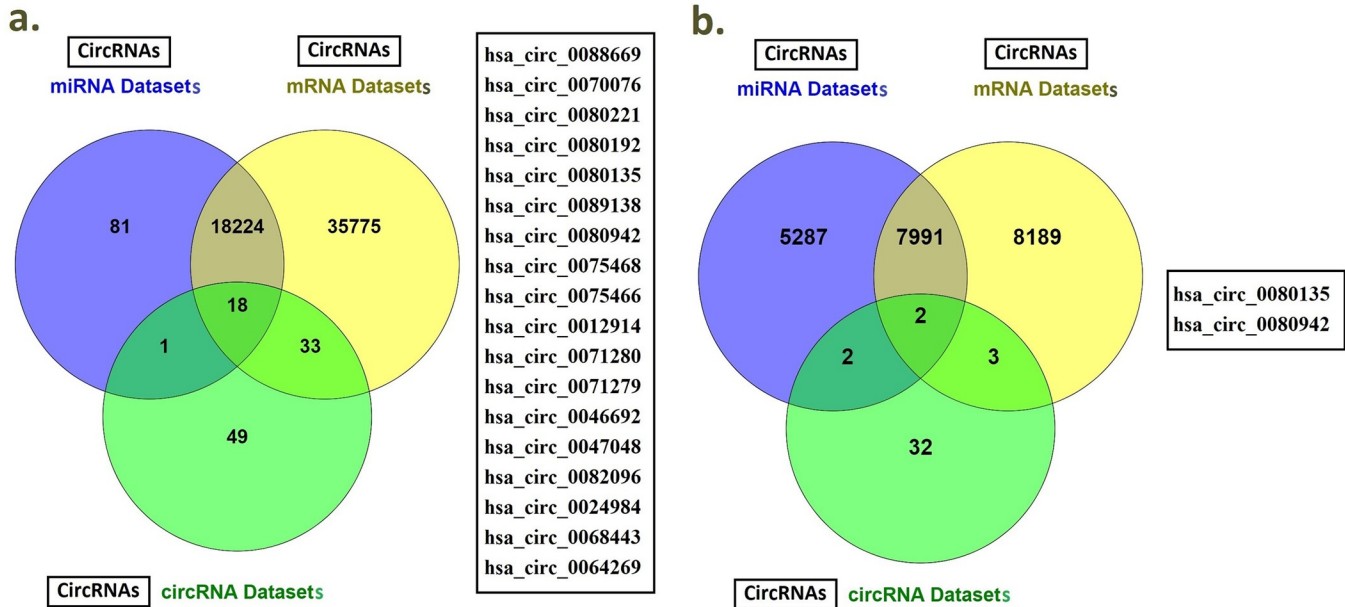

**Fig 3. Venn diagram of overlapped differentially expressed circRNAs among circRNAs of circRNA datasets, miRNA datasets and mRNA datasets.** a) Overlapped differentially expressed circRNAs among circRNAs of circRNA datasets, miRNA datasets and mRNA datasets (whole genes). b) Overlapped differentially expressed circRNAs among circRNAs of circRNA datasets, miRNA datasets and mRNA datasets (cytokine storm related mRNAs).

Following that, the target genes of differentially expressed miRNAs were retrieved from miRDB, miRWalk 2.0, miRTarBase, and TargetScan 7.0 web-servers. Results revealed 77722 target genes for differentially expressed miRNAs. Further analysis revealed 38937 circRNAs have binding sites for respective differentially expressed miRNAs.

## Differentially expressed mRNA

Fifteen different datasets of mRNA expression profiles were retrieved from the GEO database and data mining tools. Datasets includes GSE19137, GSE166552, PRJCA002617, Chi et al., 2020, Lin et al., 2020, Chen et al., 2020a, Chen et al., 2020b, Blanco-Melo et al., 2020, Liu et al., 2020, Del Valle et al., 2020, Qin et al., 2020, Yang et al., 2020, Farr et al., 2021, Dhar et al., 2021and Huang et al., 2020b. GSE19137 dataset revealed 171 dysregulated mRNAs. Another dataset PRJCA002617 revealed 30 dysregulated mRNAs. 25 dysregulated mRNAs were identified in GSE166552 dataset. Dataset of Chi et al., 2020 revealed 11 dysregulated mRNAs. Another dataset of Lin et al., 2020 revealed 6 dysregulated mRNAs. 6 dysregulated mRNAs were identified in Chen et al., 2020a dataset. Chen et al., 2020b dataset revealed 2 dysregulated mRNAs. In addition, 8 dysregulated mRNAs were identified in Blanco-Melo et al., 2020 dataset. Liu et al., 2020 dataset revealed 4 dysregulated mRNAs. Furthermore, another dataset Del Valle et al., 2020 revealed 3 dysregulated mRNAs. Qin et al., 2020 dataset revealed 4 dysregulated mRNAs. In addition Yang et al., 2020, dataset revealed 14 dysregulated mRNAs. In another dataset by Farr et al., 2021, findings revealed 3 dysregulated mRNAs. In addition, Dhar et al., 2021 dataset revealed 6 dysregulated mRNAs. Furthermore, a dataset of Huang et al., 2020b revealed 8 dysregulated mRNAs. Comprehensive analysis of fifteen different datasets revealed a total of 301 differentially expressed mRNAs.

## Target miRNAs of differentially expressed circRNAs

Target analysis of differentially expressed circRNAs was performed. GSE166552 dataset revealed a total of 4316 miRNA targets having binding sites on differentially expressed

circRNAs. PRJCA002617 dataset revealed a total of 1516 miRNAs targets having binding sites on differentially expressed circRNAs. Comprehensive analysis of two datasets revealed 5832 miRNAs targets having binding sites on differential expressed circRNAs.

## Target mRNAs of predicted miRNAs

Comprehensive analysis of 5832 miRNAs targets having binding sites on differential expressed circRNAs revealed 704000 genes having binding sites for respective miRNAs.

## Target mRNAs of differentially expressed miRNAs

mRNA target analysis of differentially expressed miRNAs in McDonald et al., 2021 dataset, Demirci et al., 2021, Farr et al., 2021 dataset, Li et al., 2020 dataset, Chow et al., 2020 dataset, and Arora et al., 2020 dataset revealed 77722 target genes.

## Target circRNAs of differentially expressed miRNAs

Comprehensive analysis of differentially expressed miRNAs in McDonald et al., 2021 dataset, Demirci et al., 2021, Farr et al., 2021 dataset, Li et al., 2020 dataset, Chow et al., 2020 dataset, and Arora et al., 2020 dataset revealed 38937 target circRNAs.

## Target miRNAs of differentially expressed mRNAs

miRNA target analysis of differentially expressed mRNAs from GSE19137, GSE166552, PRJCA002617, Chi et al., 2020, Lin et al., 2020, Chen et al., 2020a, Chen et al., 2020b, Blanco-Melo et al., 2020, Liu et al., 2020, Del Valle et al., 2020, Qin et al., 2020, Yang et al., 2020, Farr et al., 2021, Dhar et al., 2021and Huang et al., 2020b datasets revealed a total of 5109 miRNAs against differentially expressed mRNAs.

## Target CircRNAs of predicted miRNAs

Comprehensive analysis of 5109 predicted miRNAs against differentially expressed mRNAs revealed 858423 circRNAs having binding sites for respective miRNAs.

## Analysis of SARS-CoV-2 induced cytokine storm related circRNA-miRNA-mRNA Axis

Previous findings have revealed that research on therapeutic strategies which have the potential to counteract multiple cytokines and related signaling pathways involved in COVID-19 is the need of the hour to capitalize promising therapeutic approach. Here we analyzed 12 datasets in which authors have evaluated and studied cytokine expression profiles and signatures induced by SARS-CoV-2 (Table 4). We wanted to retrieve those circRNAs which can inhibit maximum number of cytokines induced by SARS-COV-2. Comprehensive analysis of datasets revealed a total of 74 differentially expressed mRNAs, all of which were found to be upregulated. After removing duplicates, 32 mRNAs were found to be dysregulated. miRNA target analysis of differentially expressed 32 mRNAs (IL-1β, IL2, IL4, IL-6, IL-7, IL-8, IL10, IL-12, IL-13, IL-17, IL-18, IL23, IL33, IL-37, IL-38, TNF-α, IFN-γ, CCL2, CXCL6, CXCL8, CXCL10, IP-10, MIP-1A, MIP1-B, PDGF, MCP1, GM-CSF, M-CSF, G-CSF, FGF, HGF, TGF-β) revealed a total of 262 miRNAs against differentially expressed mRNAs. Comprehensive analysis of 262 predicted miRNAs against differentially expressed 32 mRNAs revealed 99419 circRNAs having binding sites for respective miRNAs.

**Table 4. Datasets used for the analysis of SARS-CoV-2 related cytokines.**

| Genes | Study 1 up-regulated | Study 1 down-regulated | Study 2 up-regulated | Study 2 down-regulated | Study 3 up-regulated | Study 3 down-regulated | Study 4 up-regulated | Study 4 down-regulated | Study 5 up-regulated | Study 5 down-regulated | Study 6 up-regulated | Study 6 down-regulated | Study 7 up-regulated | Study 7 down-regulated | Study 8 up-regulated | Study 8 down-regulated | Study 9 up-regulated | Study 9 down-regulated | Study 10 up-regulated | Study 10 down-regulated | Study 11 up-regulated | Study 11 down-regulated | Study 12 up-regulated | Study 12 down-regulated |
|---|---|---|---|---|---|---|---|---|---|---|---|---|---|---|---|---|---|---|---|---|---|---|---|---|
| IL-1β | – | – | – | – | – | – | – | – | ✓ | – | – | – | ✓ | – | – | – | – | – | – | – | – | – | – | – |
| IL-2 | ✓ | – | – | – | – | – | – | – | – | – | – | – | ✓ | – | – | – | – | – | – | – | ✓ | – | ✓ | – |
| IL-4 | – | – | – | – | – | – | – | – | – | – | – | – | ✓ | – | – | – | – | – | – | – | – | – | ✓ | – |
| IL-6 | – | – | ✓ | – | ✓ | – | – | – | ✓ | – | ✓ | – | ✓ | – | ✓ | – | ✓ | – | ✓ | – | ✓ | – | ✓ | – |
| IL-7 | ✓ | – | – | – | – | – | – | – | – | – | – | – | – | – | – | – | – | – | ✓ | – | – | – | – | – |
| IL-8 | – | – | – | – | – | – | – | – | ✓ | – | ✓ | – | – | – | ✓ | – | – | – | ✓ | – | – | – | – | – |
| IL-10 | ✓ | – | – | – | ✓ | – | – | – | ✓ | – | – | – | ✓ | – | – | – | – | – | ✓ | – | – | – | ✓ | – |
| IL-12 | – | – | – | – | – | – | – | – | – | – | – | – | – | – | – | – | – | – | – | – | – | – | – | – |
| IL-13 | – | – | – | – | – | – | – | – | – | – | – | – | – | – | – | – | – | – | – | – | ✓ | – | – | – |
| IL-17 | – | – | – | – | – | – | – | – | – | – | – | – | – | – | – | – | – | – | – | – | – | – | – | – |
| IL-18 | – | – | – | – | – | – | – | – | – | – | – | – | – | – | – | – | – | – | ✓ | – | ✓ | – | – | – |
| IL-23 | – | – | – | – | – | – | – | – | – | – | – | – | – | – | – | – | – | – | – | – | – | – | – | – |
| IL-33 | – | – | – | – | – | – | – | – | – | – | – | – | – | – | – | – | – | – | – | – | – | – | – | – |
| IL-37 | – | – | – | – | – | – | – | – | – | – | – | – | – | – | – | – | – | – | – | – | – | – | – | – |
| IL-38 | – | – | – | – | – | – | – | – | – | – | – | – | – | – | – | – | – | – | – | – | – | – | – | – |
| TNF-α | ✓ | – | – | – | ✓ | – | – | – | ✓ | – | ✓ | – | ✓ | – | ✓ | – | – | – | – | – | – | – | ✓ | – |
| IFN-γ | – | – | – | – | ✓ | – | – | – | – | – | – | – | ✓ | – | – | – | – | – | – | – | – | – | ✓ | – |
| CCL2 | – | – | – | – | – | – | – | – | – | – | – | – | – | – | – | – | ✓ | – | – | – | – | – | – | – |
| CXCL6 | – | – | – | – | – | – | – | – | – | – | – | – | – | – | – | – | – | – | – | – | – | – | – | – |
| CXCL8 | – | – | – | – | – | – | – | – | – | – | – | – | – | – | – | – | ✓ | – | – | – | – | – | – | – |
| CXCL10 | – | – | – | – | – | – | – | – | – | – | – | – | – | – | – | – | ✓ | – | – | – | – | – | – | – |
| IP-10 | ✓ | – | – | – | – | – | ✓ | – | – | – | – | – | – | – | – | – | – | – | ✓ | – | ✓ | – | – | – |
| MIP-1A | ✓ | – | – | – | – | – | ✓ | – | – | – | – | – | – | – | – | – | – | – | ✓ | – | ✓ | – | – | – |
| MIP1-B | – | – | – | – | – | – | ✓ | – | – | – | – | – | – | – | – | – | – | – | – | – | ✓ | – | – | – |
| PDGF | – | – | – | – | – | – | – | – | – | – | – | – | – | – | – | – | – | – | – | – | ✓ | – | – | – |
| MCP1 | ✓ | – | – | – | – | – | ✓ | – | – | – | – | – | – | – | – | – | – | – | ✓ | – | – | – | – | – |
| GM-CSF | – | – | – | – | – | – | – | – | – | – | – | – | – | – | – | – | – | – | – | – | – | – | – | – |
| M-CSF | – | – | – | – | – | – | – | – | – | – | – | – | – | – | – | – | – | – | ✓ | – | ✓ | – | – | – |
| G-CSF | ✓ | – | – | – | – | – | – | – | – | – | – | – | – | – | – | – | – | – | ✓ | – | ✓ | – | – | – |
| FGF | – | – | – | – | – | – | – | – | – | – | – | – | – | – | – | – | – | – | – | – | – | – | – | – |
| HGF | – | – | – | – | – | – | – | – | – | – | – | – | – | – | – | – | – | – | – | – | ✓ | – | – | – |
| TGF-β | – | – | – | – | – | – | – | – | – | – | – | – | – | – | – | – | – | – | – | – | – | – | – | – |
|  | [45] | | [39] | | [38] | | [46] | | [42] | | [32] | | [44] | | [41] | | [40] | | [36] | | [43] | | [37] | |

## Common circRNA-miRNA-mRNA pairs among circRNA, miRNA and mRNA datasets

Analysis of common circRNAs among circRNA, miRNA and mRNA datasets revealed two circRNAs including hsa_circ_0080942 and hsa_circ_0080135 (Fig 3). The two filtered circRNAs have binding sites for 86 miRNAs. These 86 miRNAs have binding sites for 15 cytokine storm related mRNAs including IL-1β, IL-7, IL-10, IL-12B, IL-13, IL-17A, IL-33, IFN-γ, CCL2, CXCL6, CXCL8, CXCL10, MIP, FGF2, FGF14.

## Construction of circRNA-miRNA-mRNA networks

The circRNA-miRNA-mRNA network of hsa_circ_0080942 and hsa_circ_0080135 was visualized using the Cytoscape software. The hsa_circ_0080942 network included 47 miRNAs forming 88 pairs of circRNA-miRNA-mRNA axis. For instance, hsa_circ_0080942 is the ceRNA of hsa-miR-1183 targeting MIP, CXCL8, CXCL10, IL33, IL1B, IL10, IL17A, IL12B. Furthermore, hsa_circ_0080942 is the ceRNA of hsa-miR-486-3p targeting IL33, IL13, IL1B and IL7. The hsa_circ_0080135 network included 39 miRNAs forming 77 pairs of circRNA-miRNA-mRNA axis. For instance, hsa_circ_0080135 is the ceRNA of hsa-miR-885-3p targeting IL12B, IL1B, IL33, CXCL10 and MIP. Furthermore, hsa_circ_0080135 is the ceRNA of hsa-miR-769-3p targeting IL12B, IFNG, CXCL6 and CXCL8 (Fig 4) (Table 5).

## KEGG pathway and functional enrichment analysis

DAVID was used to conduct a KEGG pathway enrichment analysis to gain a better understanding of the differentially expressed genes' function (Fig 5) (Table 6). In order to screen

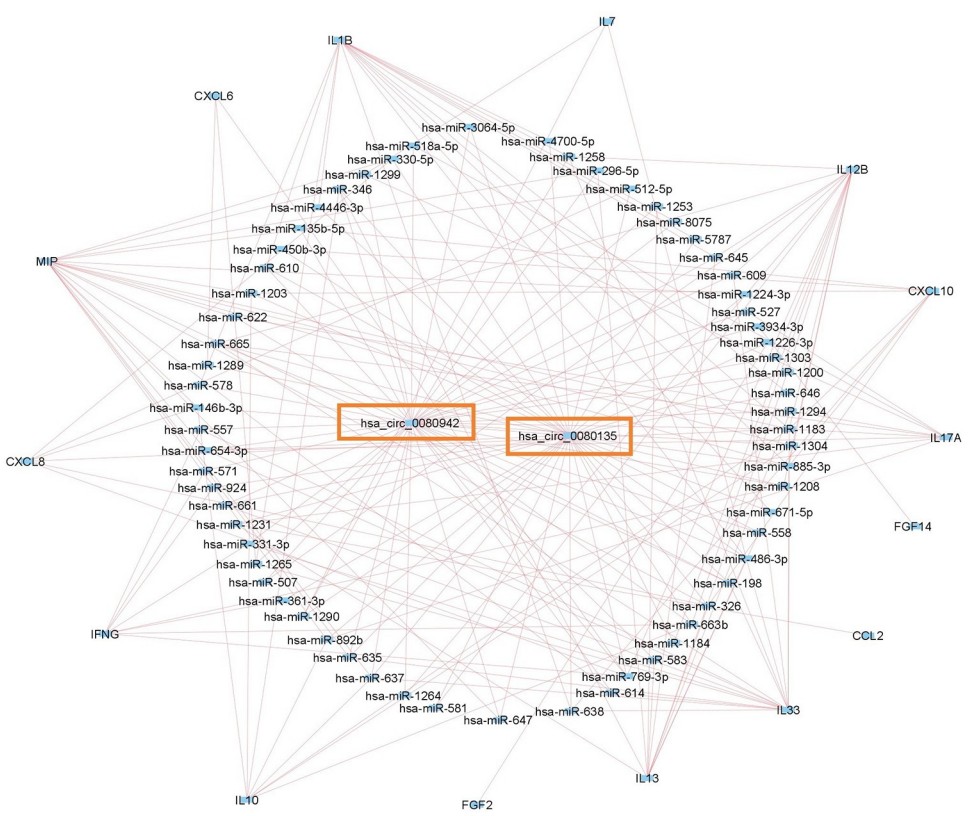

**Fig 4. The SARS-CoV-2 induced cytokine storm related circRNA-miRNA-mRNA network visualized using Cytoscape software.** The circRNA-miRNA-mRNA network contains 81 nodes and 205 edges.

**Table 5. SARS-CoV-2 induced cytokine storm related circRNA-miRNA-mRNA regulatory axis.**

| CircRNA | Parent Gene | miRNA | mRNA |
|---|---|---|---|
| **hsa_circ_0080942** | PCLO | hsa-miR-1183 | MIP/CXCL8/CXCL10/IL33/IL1B/IL10/IL17A/IL12B |
| | | hsa-miR-486-3p | IL33/IL13/IL1B/IL7 |
| | | hsa-miR-769-3p | IL12B/IFNG/CXCL6/CXCL8 |
| | | hsa-miR-637 | IL33/IL10/MIP |
| | | hsa-miR-1265 | IL1B/IL33/CXCL6 |
| | | hsa-miR-654-3p | IL17A/IL33/MIP |
| | | hsa-miR-665 | IL17A/IL33/IL12B |
| | | hsa-miR-346 | IL13/IL10/MIP |
| | | hsa-miR-1258 | IL12B/IL1B/IL33 |
| | | hsa-miR-645 | IL12B/IL13/IL1B |
| | | hsa-miR-1303 | IL12B/IL10/MIP |
| | | hsa-miR-198 | IL33/MIP |
| | | hsa-miR-614 | IL33/MIP |
| | | hsa-miR-635 | IL33/MIP |
| | | hsa-miR-331-3p | IL33/IFNG |
| | | hsa-miR-557 | IL33/IFNG |
| | | hsa-miR-622 | IL17A/MIP |
| | | hsa-miR-1299 | IL17A/IL7 |
| | | hsa-miR-296-5p | IL13/MIP |
| | | hsa-miR-609 | IL12B/IL33 |
| | | hsa-miR-1200 | IL12B/IL13 |
| | | hsa-miR-1208 | IL12B/IFNG |
| | | hsa-miR-663b | IL10/IFNG |
| | | hsa-miR-1224-3p | MIP |
| | | hsa-miR-513a-5p | MIP |
| | | hsa-miR-636 | MIP |
| | | hsa-miR-647 | MIP |
| | | hsa-miR-1290 | IL7 |
| | | hsa-miR-661 | IL33 |
| | | hsa-miR-578 | IL1B |
| | | hsa-miR-610 | IL1B |
| | | hsa-miR-671-5p | IL1B |
| | | hsa-miR-518a-5p | IL17A |
| | | hsa-miR-527 | IL17A |
| | | hsa-miR-1253 | IL13 |
| | | hsa-miR-1294 | IL13 |
| | | hsa-miR-558 | IL13 |
| | | hsa-miR-583 | IL13 |
| | | hsa-miR-1264 | IL12B |
| | | hsa-miR-507 | IL12B |
| | | hsa-miR-571 | IFNG |
| | | hsa-miR-924 | IFNG |
| | | hsa-miR-1289 | CXCL6 |
| | | hsa-miR-450b-3p | CXCL10 |
| | | hsa-miR-135b-5p | CXCL10 |
| | | hsa-miR-3934-3p | FGF14 |
| | | hsa-miR-3064-5p | IL1B |

*(Continued)*

**Table 5.** (Continued)

| CircRNA | Parent Gene | miRNA | mRNA |
|---|---|---|---|
| **hsa_circ_0080135** | TNS3 | hsa-miR-885-3p | IL12B/IL1B/IL33/CXCL10/MIP |
| | | hsa-miR-486-3p | IL33/IL13/IL1B/IL7 |
| | | hsa-miR-326 | IL13/CXCL8/CXCL10/MIP |
| | | hsa-miR-769-3p | IL12B/IFNG/CXCL6/CXCL8 |
| | | hsa-miR-637 | IL33/IL10/MIP |
| | | hsa-miR-638 | IL1B/IL33/CXCL10 |
| | | hsa-miR-654-3p | IL17A/IL33/MIP |
| | | hsa-miR-665 | IL17A/IL33/IL12B |
| | | hsa-miR-892b | IL17A/IL10/MIP |
| | | hsa-miR-1231 | IL13/IL17A/CCL2 |
| | | hsa-miR-346 | IL13/IL10/MIP |
| | | hsa-miR-146b-3p | IL10/CXCL8/MIP |
| | | hsa-miR-1203 | IL10/CXCL10/MIP |
| | | hsa-miR-198 | IL33/MIP |
| | | hsa-miR-635 | IL33/MIP |
| | | hsa-miR-1200 | IL12B/IL13 |
| | | hsa-miR-1208 | IL12B/IFNG |
| | | hsa-miR-663b | IL10/IFNG |
| | | hsa-miR-330-5p | CXCL8/MIP |
| | | hsa-miR-512-5p | CXCL8/MIP |
| | | hsa-miR-1224-3p | MIP |
| | | hsa-miR-647 | MIP |
| | | hsa-miR-1290 | IL7 |
| | | hsa-miR-661 | IL33 |
| | | hsa-miR-610 | IL1B |
| | | hsa-miR-646 | IL1B |
| | | hsa-miR-671-5p | IL1B |
| | | hsa-miR-558 | IL13 |
| | | hsa-miR-1184 | IL12B |
| | | hsa-miR-581 | IL12B |
| | | hsa-miR-361-3p | IFNG |
| | | hsa-miR-924 | IFNG |
| | | hsa-miR-1289 | CXCL6 |
| | | hsa-miR-450b-3p | CXCL10 |
| | | hsa-miR-3064-5p | IL1B |
| | | hsa-miR-4446-3p | IL1B |
| | | hsa-miR-4700-5p | IL1B |
| | | hsa-miR-5787 | IL1B |
| | | hsa-miR-1226-3p | FGF2 |

core and hub genes as well as physical and functional associations, 203 differentially expressed genes were imported to STRING web-based tool. These 203 differentially expressed genes represent the results of the mRNA datasets analyzed in the current manuscript. The threshold condition was a confidence score ≥0.4. The results of analyzed data were then imported into Cytoscape for visualization analysis, and node connectivity was calculated to screen for the central node (hub genes) of the network. Finally the relationship between hub genes and prioritized circRNAs were determined and analyzed. There were 203 nodes and 657 edges in the

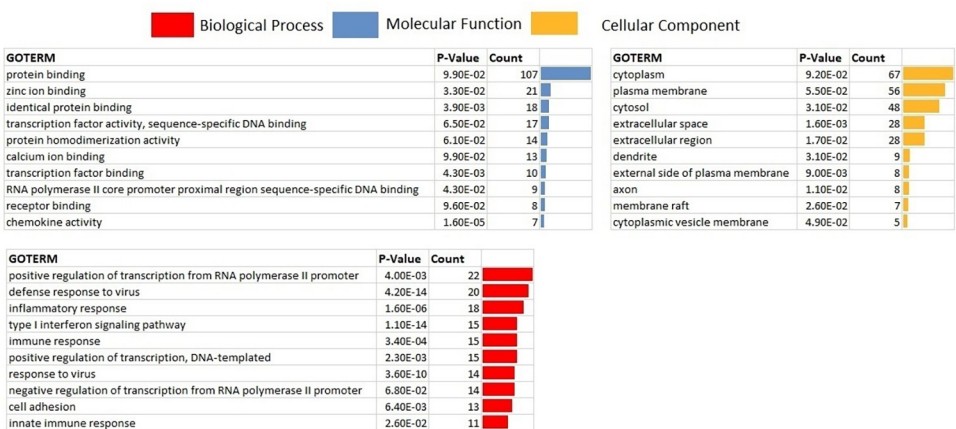

**Fig 5. Gene ontology analysis of differentially expressed genes.** Top GO terms with lowest P-values in cellular component, molecular function, and biological process were shown, respectively.

PPI network. The cytoHubba plug-in in Cytoscape was used to search the list of top 10 genes from the PPI network with node degrees indicating hub DEGs, including STAT1, RSAD2, IFIT1, IFIT3, IFIT2, DDX58, OAS2, MX2, IFI44 and IFI44L (Fig 6). Module analysis using MCODE plug in revealed 6 modules; module 1 containing 25 nodes and 272 edges, module 2 containing 6 nodes and 11 edges, module 3 containing 5 nodes and 7 edges, module 4 containing 5 nodes and 3 edges, module 5 containing 3 nodes and 3 edges and module 6 containing 3 nodes and 3 edges.

## Pharmacogenomics analysis for hub genes

By exploring the website PharmGkb, genes indirectly targeted by two prioritized circRNAs were selected for Pharmacogenomics analysis to find some potential drugs. The results revealed that HMG-CoA reductase inhibitors (target: STAT1), simvastatin (target: STAT1), atorvastatin (target: STAT1), aspirin (target: STAT1), pravastatin (target: STAT1), rosuvastatin (target: STAT1), clomipramine (target: MIP), Imipramine (target: MIP), Mercaptopurine (target: MIP), trimipramine (target: MIP), diosmectite (target: CXCL10), canakinumab (target: IL-1B), anakinra (target: IL-1B), secukinumab (target: IL-17A), ixekizumab (target: IL-17A), and chloroquine (FGF2) might serve as potential therapeutic options for SARS-CoV-2 infection (Table 7). Many of these drugs have been approved by FDA against COVID-19 and previously been found to be effective in mitigating the effect of SARS-CoV-2, further increasing the

**Table 6. KEGG analysis of 15 cytokine storm related genes.**

| Sr. No | KEGG Pathway | Enrichment Score | P value | Cytokines Count |
|---|---|---|---|---|
| 1 | Chemokine signaling pathway | 4.33 | 4.6E-3 | 4 |
| 2 | Inflammatory bowel disease (IBD) | 3.82 | 7.2E-8 | 6 |
| 3 | Rheumatoid arthritis | 3.82 | 3.6E-7 | 6 |
| 4 | Toll-like receptor signaling pathway | 3.82 | 9.1E-4 | 4 |
| 5 | NOD-like receptor signaling pathway | 3.82 | 4.8E-3 | 3 |
| 6 | TNF signaling pathway | 3.44 | 1.7E-2 | 3 |
| 7 | RIG-I-like receptor signaling pathway | 3.44 | 7.4E-3 | 3 |
| 8 | Herpes simplex infection | 3.44 | 4.4E-3 | 4 |
| 9 | Influenza A | 3.44 | 3.6E-7 | 7 |

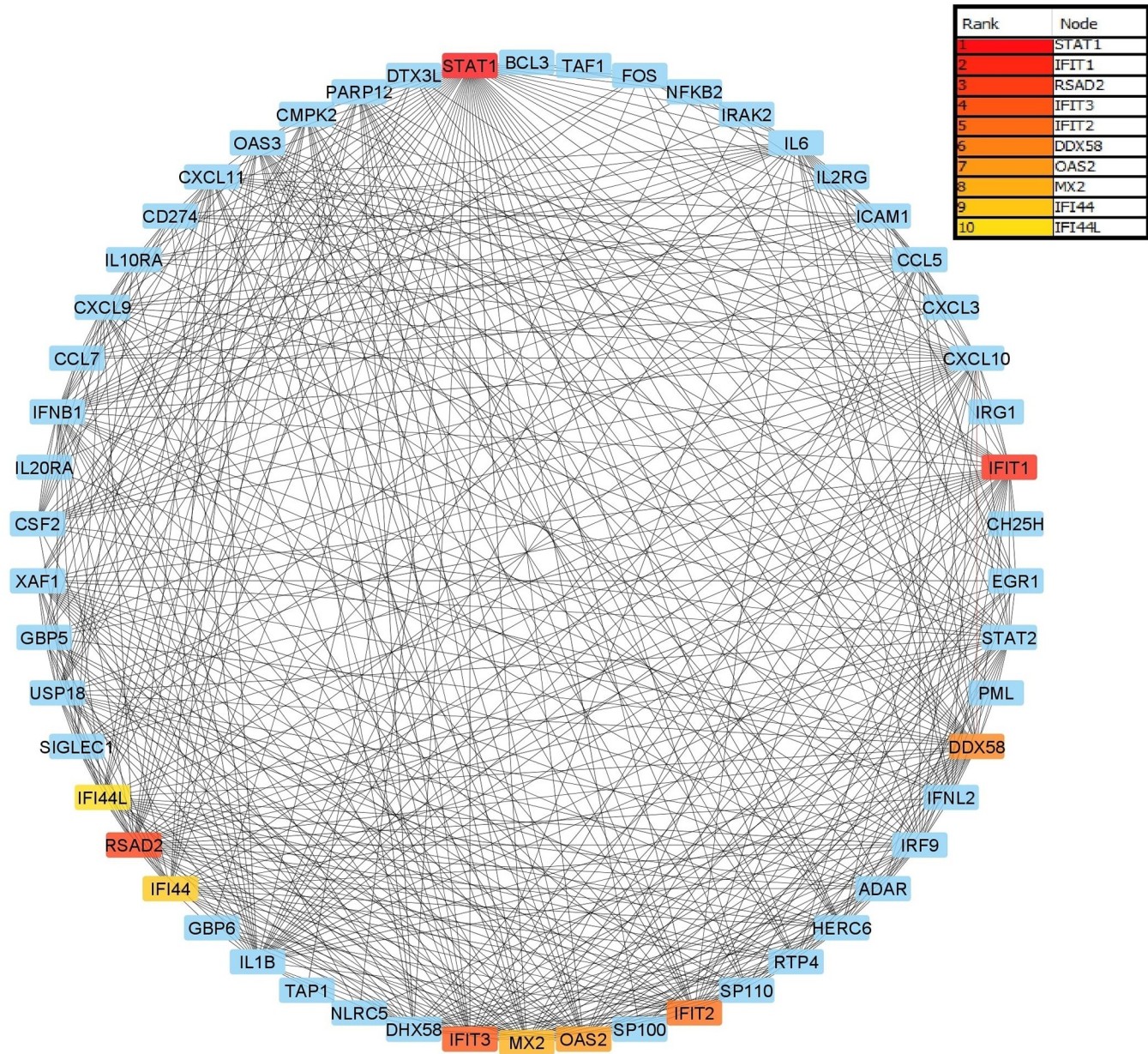

| Rank | Node |
|------|--------|
| 1 | STAT1 |
| 2 | IFIT1 |
| 3 | RSAD2 |
| 4 | IFIT3 |
| 5 | IFIT2 |
| 6 | DDX58 |
| 7 | OAS2 |
| 8 | MX2 |
| 9 | IFI44 |
| 10 | IFI44L |

**Fig 6. The cytoHubba plug-in in Cytoscape was used to search the list of top 10 genes from the PPI network with node degrees indicating hub differentially expressed genes, including STAT1, RSAD2, IFIT1, IFIT3, IFIT2, DDX58, OAS2, MX2, IFI44 and IFI44L.**

reliability of current study results. Study of Bergqvist et al., revealed negative association between Statin treatment and COVID-19 mortality [66]. Mercaptopurine, a purine analogue has been previously reported as a selective inhibitor of both MERS-CoV and SARS-CoV [67]. In another study, authors found that a tricyclic antidepressant trimipramine protected against SARS-CoV-2 induced cytopathic effect via inhibiting autophagy [68]. Moreover, Clomipramine was found to be effective in preventing neurological manifestations of SARS-CoV-2 Infection [69]. In a retrospective study, statins were found to have potential beneficial effects on mortality rates associated with SARS-CoV-2 infection [70]. Diosmectite, which is an aluminomagnesium silicate adsorbent clay has been recommended for managing COVID-19.

**Table 7. Drug targets and their association with prioritized circRNAs during SARS-CoV-2 infection.**

| Drugs | mRNAs | miRNAs | CircRNAs |
|---|---|---|---|
| **Clomipramine Imipramine Mercaptopurine Trimipramine** | MIP | hsa-miR-1183 | hsa_circ_0080942 |
| | | hsa-miR-637 | |
| | | hsa-miR-654-3p | |
| | | hsa-miR-346 | |
| | | hsa-miR-1303 | |
| | | hsa-miR-198 | |
| | | hsa-miR-614 | |
| | | hsa-miR-635 | |
| | | hsa-miR-622 | |
| | | hsa-miR-296-5p | |
| | | hsa-miR-1224-3p | |
| | | hsa-miR-513a-5p | |
| | | hsa-miR-636 | |
| | | hsa-miR-647 | |
| **hmg coa reductase inhibitors** **simvastatin** **atorvastatin** **aspirin** **pravastatin** **rosuvastatin** | STAT1 | | |
| **Diosmectite** | CXCL10 | hsa-miR-1183 | hsa_circ_0080942 |
| | | hsa-miR-450b-3p | |
| | | hsa-miR-135b-5p | |
| **Canakinumab Anakinra** | IL-1 B | hsa-miR-1183 | hsa_circ_0080942 |
| | | hsa-miR-486-3p | |
| | | hsa-miR-1265 | |
| | | hsa-miR-1258 | |
| | | hsa-miR-645 | |
| | | hsa-miR-578 | |
| | | hsa-miR-610 | |
| | | hsa-miR-671-5p | |
| | | hsa-miR-3064-5p | |
| **Secukinumab Ixekizumab** | IL-17A | hsa-miR-1183 | hsa_circ_0080942 |
| | | hsa-miR-486-3p | |
| | | hsa-miR-654-3p | |
| | | hsa-miR-665 | |
| | | hsa-miR-622 | |
| | | hsa-miR-1299 | |
| | | hsa-miR-518a-5p | |
| | | hsa-miR-527 | |
| **Chloroquine** | FGF2 | hsa-miR-1226-3p | hsa_circ_0080135 |

Exposure of this drug inhibited CXCL10 secretion and NF-kappaB activation [71]. Moreover both anakinra (by blocking IL-1 receptor) and canakinumab (by blocking the IL-1 signaling) can potentially interrupt autoinflammatory loop during SARS-CoV-2 infection. Another drug Chloroquine has been reported to inhibit FGF2-induced mitogenic activity [72].

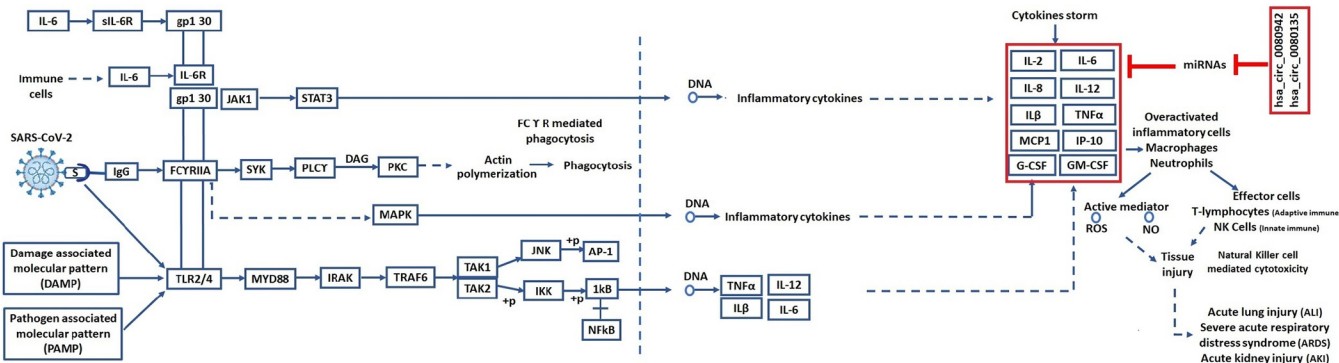

**Fig 7. Pathway analysis of COVID-19 pathogenesis (KEGG pathway ID: map05171).** Highlighted genes are targets of miRNAs and indirect targets of two prioritized circRNAs.

Multiple drugs approved right now for the treatment of SARS-CoV2 infection have targets been regulated by two prioritized circRNAs. Instead of using multiple drugs targeting multiple mRNAs, we can utilize single circRNA regulating multiple disease related mRNAs. We have also retrieved the Coronavirus disease-COVID-19 pathway from KEGG database with pathway ID: map05171. This pathway illustrates those cytokines that are part of cytokine syndrome induced during the SARS-CoV-2 infection. Further we have highlighted the target cytokines of prioritized miRNAs and circRNAs of current study to decipher the importance of future therapeutic avenues using circRNAs (Fig 7).

## Discussion

A number of studies have highlighted significant involvement of circRNA-miRNA-mRNA regulatory axis in signaling pathways of human diseases [24]. CircRNA, a type of highly conserved endogenous RNA, has been shown to operate as a "sponge" that absorbs matching miRNA by engaging with miRNA binding sites and therefore indirectly regulates gene expression [73]. In the current approach (Fig 2), we have systematically analyzed datasets to explore circRNA, miRNA, and mRNA expression profiles during SARS-CoV-2 infection. Findings of our study revealed differential expression profiles of circRNAs, mRNAs and miRNAs during SARS-CoV-2 infection. Functional analysis was performed along with construction of protein-protein interaction network and circRNA-miRNA-mRNA network. Final circRNA-miRNA-mRNA network was constructed based on cytokine storm related circRNAs forming a total of 165 circRNA-miRNA-mRNA pairs. Our investigation of the triple regulatory networks of circRNA-miRNA-mRNA revealed two circRNAs including hsa_circ_0080942 and hsa_circ_0080135 as a potential theranostic agents for SARS-CoV-2 infection. Further these results have shed light on the complex regulatory mechanism of circRNAs in SARS-CoV-2.

Analysis of miRNA targets of prioritized circRNAs revealed significant role of respective miRNAs in SARS-CoV-2 pathogenesis. In one of the study, the expression level of hsa-miR-135b-5 was found to be down-regulated in lung epithelial cells infected with SARS-CoV-2. The authors concluded that, lowering the expression level of host miRNAs is expected to make the respiratory epithelium more susceptible to infection where as increasing the expression levels of host miRNAs might mitigate coronavirus infection [34]. In another study, miR-8066 was found to be associated with cytokine storm which is one of the major COVID-19 problem [74]. Authors revealed that miR-8066 significantly impacted cytokine-cytokine receptor pathway. Moreover, miR-3934-3p have been found to down-regulate TGFBR1 and SMAD3 which are critical players for lung fibrosis and have been previously reported in SARS-CoV-related

cases [74]. The work of Peng et al., revealed regulatory role of miR-486-5p in influenza virus replication. The authors proposed that designing miRNA-based therapies against viral infection might be a useful strategy as increased expression of miRNAs elicits effective antiviral defenses against influenza A viruses [75]. Sang et al., revealed that down-regulation of miR-637 was the causative factor of Pulmonary hypertension [76]. In another work, the authors found that suppressing the level of cellular miRNA hsa-miR-1258 enhanced viral particle production by 2.99 folds [77]. Another study revealed that Coxsackievirus A16 induced down-regulation of miR-1303 promoted the disruption of blood brain barrier integrity via miR-1303-MMP axis [78]. The work performed by Sung et al., revealed anti-HIV-1 activity of miR-198 via targeting cyclin T1 [79]. Bagasra et al., found down-regulation of has-miR-5787 in Zika virus infected neuronal cell line [80]. A number of other studies have revealed down-regulation of miRNA targets of prioritized circRNAs in viral infections and other diseases. Targeting circRNAs as a therapeutic approach might seems feasible strategy as single circRNA can target multiple miRNAs.

Further we also analyzed hub genes and cytokine storm related genes of prioritized circRNAs. Orumaa and co-authors revealed that severity of COVID-19 is linked to cytokine storm, which occurs when levels of inflammatory mediators such as IL-7, IL-10, and MIP are up-regulated [81]. Hassan et al., revealed the connection of IL-17 to critically sick patients infected with SARS-CoV-2 [82]. [83] study showed that over-expression of IL1B could be impacted by up-regulation of IRF8 and MYD88 [83]. The pro-inflammatory cytokine IL1B, which encodes IL-1, is one of the key mediators in inducing innate immune response-mediated inflammation in COVID-19 patients' lungs. In one of the study, Gupta et al., showed that activation of the cytokines including CXCL6, CXCL8 and CXCL10 is a distinctive profile of cytokine response in COVID-19 patients [84]. In another study, authors found association between increased level of FGF2 and patients with severe COVID-19 [85]. The work performed by Donlan et al., discovered that IL-13 levels in severe COVID-19 patients requiring ICU and/or mechanical ventilation rose from day 5 to day 20 of illness. These findings suggest that IL-13 is a key component of the host's response to SARS-CoV-2 infection and may be driving factor for severe illness [86]. A previous study found that seriously affected COVID-19 patients had an increased level of STAT1 and IRF9 [87]. Work of Giovannoni and coauthors revealed increased expression levels of RSAD2 in COVID-19 and found substantial correlation of RSAD2 with viral load [88]. Moreover another study revealed increased level of DDX58 in COVID-19-positive patients [89]. Another study found increased levels of IFI27 and OAS2 in COVID-19 patients [90].

Integrative analysis performed by Lu and coauthors revealed regulatory activities of circRNAs and their particular interaction with other RNAs via circRNA-miRNA-mRNA regulatory axis during Hantavirus infection [91]. In addition various recent study showed that circRNAs have been shown to bind to miRNAs and serve as natural miRNA sponges, influencing the activities of associated miRNAs and the gene expressions controlled by miRNAs [92]. Prioritized circRNAs includes hsa_circ_0080942 and hsa_circ_0080135. The hsa_circ_0080942 network included 47 miRNAs and 15 cytokine storm related genes forming 88 pairs of circRNA-miRNA-mRNA axis. For instance, hsa_circ_0080942 is the ceRNA of hsa-miR-1183 targeting MIP, CXCL8, CXCL10, IL33, IL1B, IL10, IL17A, IL12B. Furthermore, hsa_circ_0080942 is the ceRNA of hsa-miR-486-3p targeting IL33, IL13, IL1B and IL7. The hsa_circ_0080135 network included 39 miRNAs and 15 cytokine storm related genes forming 77 pairs of circRNA-miRNA-mRNA axis. For instance, hsa_circ_0080135 is the ceRNA of hsa-miR-885-3p targeting IL12B, IL1B, IL33, CXCL10 and MIP. Furthermore, hsa_circ_0080135 is the ceRNA of hsa-miR-769-3p targeting IL12B, IFNG, CXCL6 and CXCL8.

Via targeting two prioritized circRNAs, maximum number of cytokines dysregulated during SARS-CoV-2 infection can be targeted.

Furthermore Pharmacogenomics analysis was performed to retrieve some potentials therapeutic options for SARS-CoV-2 infection. Many of these drugs were previously found to be effective in mitigating the effect of SARS-CoV-2. Study of Bergqvist et al., revealed negative association between Statin treatment and COVID-19 mortality [66]. Mercaptopurine, a purine analogue has been previously reported as a selective inhibitor of both MERS-CoV and SARS-CoV [67]. In another study, authors found that a tricyclic antidepressant trimipramine protected against SARS-CoV-2 induced cytopathic effect via inhibiting autophagy [68]. Moreover, Clomipramine was found to be effective in preventing neurological manifestations of SARS-CoV-2 Infection [69]. In a retrospective study, statins were found to have potential beneficial effects on mortality rates associated with SARS-CoV-2 infection [70].

Another important consideration is that there are variety of conceptually different approaches for inferring and analyzing these datasets. Many comparisons have been conducted to determine which one is most suitable and reliable. The problem is that the results of such technical comparisons depend crucially on the studied conditions, including; type of the data, number of samples, amount of noise, experimental design, type of the underlying interaction structure and how you are measuring the error. For this reason it is unlikely that there is one "right" method that fits all different biological, technical and experimental design conditions best. It is also highly unlikely that there is just one method that outperforms all others for all conditions. In the current study, the construction of circRNA/miRNA/mRNA regulatory networks and the prediction of therapeutic drugs were all relying on a series of bioinformatics algorithms and databases. Still a large number of experiments are needed to verify the accuracy of these prediction conclusions. In addition, in the choice of differentially expressed RNAs, the selection criteria with higher credibility are adopted. Most reliable and reasonable algorithms were selected, but random errors and selection bias cannot be avoided.

Previous studies have indicated that circRNA is a type of high-efficiency competing endogenous RNA (ceRNA) and operate as part of ceRNA regulatory networks. It can regulate the expression level of multiple target genes by exerting a miRNA sequestering effect. This work along with other recent studies suggests that by inhibiting disease related circRNAs might present a new generation of versatile and adjustable RNA therapeutics with significant potential.

Ongoing clinical trials for the treatment of COVID-19 have proposed different therapeutic options including various drugs, monoclonal antibodies, immunoglobulin therapy, convalescent plasma therapy and cell therapy. However, at present, effects, safety, and efficacy of current treatment strategies are still uncertain and therefore more prospective clinical studies are needed in the future for further evaluation [93, 94]. Research on therapeutic strategies which have the potential to counteract multiple cytokines and related signaling pathways involved in COVID-19 is the need of the hour to capitalize promising therapeutic approach. Targeting prioritized circRNAs might provide attractive treatment option. Since circRNA is an upstream regulator of miRNA and mRNA, it may be possible to develop a circRNA–miRNA–mRNA panel, possibly including the hsa_circ_0080942 and hsa_circ_0080135 described in this work, for clinical applications as potential theranostic agents in SARS-CoV-2 infection.

## Author Contributions

**Conceptualization:** Faryal Mehwish Awan, Burton B. Yang.

**Data curation:** Hassan Ayaz, Nouman Aslam, Badr Alzahrani, Muhammad Arif.

**Formal analysis:** Hassan Ayaz, Nouman Aslam, Faryal Mehwish Awan, Rabea Basri.

**Funding acquisition:** Badr Alzahrani, Sadiq Noor Khan.

**Methodology:** Hassan Ayaz, Nouman Aslam, Faryal Mehwish Awan, Rabea Basri, Bisma Rauff, Badr Alzahrani, Aqsa Ikram, Ayesha Obaid, Anam Naz, Azhar Nazir.

**Resources:** Sadiq Noor Khan.

**Supervision:** Faryal Mehwish Awan.

**Validation:** Faryal Mehwish Awan, Burton B. Yang.

**Visualization:** Rabea Basri, Bisma Rauff, Badr Alzahrani, Muhammad Arif, Aqsa Ikram, Ayesha Obaid, Anam Naz.

**Writing – original draft:** Hassan Ayaz, Nouman Aslam, Faryal Mehwish Awan, Rabea Basri, Bisma Rauff, Muhammad Arif, Aqsa Ikram, Ayesha Obaid, Anam Naz, Azhar Nazir.

**Writing – review & editing:** Faryal Mehwish Awan, Sadiq Noor Khan, Burton B. Yang.

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
