## [Decision Letter · Decision Letter 0]

10 Jan 2023

PONE-D-22-34657Mapping CircRNA–miRNA–mRNA Regulatory Axis Identifies hsa_circ_0080942 and hsa_circ_0080135 as a potential theranostic agents for SARS-CoV-2 infectionPLOS ONE

Dear Dr. Awan,

Thank you for submitting your manuscript to PLOS ONE. After careful consideration, we feel that it has merit but does not fully meet PLOS ONE’s publication criteria as it currently stands. Therefore, we invite you to submit a revised version of the manuscript that addresses the points raised during the review process.

We look forward to receiving your revised manuscript.

Kind regards,

Kanhaiya Singh, Ph.D

Academic Editor

PLOS ONE

Journal Requirements:

2. We noted in your submission details that a portion of your manuscript may have been presented or published elsewhere. [All the datasets analyzed during the current study are accessible from the literature as well as from the GEO database (https://www.ncbi.nlm.nih.gov/gds/) with accession details (GSE166552, GSE19137, PRJCA002617, (McDonald, Enguita et al. 2021), (Arora, Singh et al. 2020), (Farr, Rootes et al. 2021), (Li, Hu et al. 2020), (Chow and Salmena 2020), (Demirci and Demirci 2021), (Chi, Ge et al. 2020), (Lin, Luo et al. 2020), (Chen, Wu et al. 2020), (Chen, Liu et al. 2020), (Blanco-Melo, Nilsson-Payant et al. 2020), (Li, Hu et al. 2020), (Del Valle, Kim-Schulze et al. 2020), (Qin, Zhou et al. 2020), (Yang, Shen et al. 2020), (Dhar, Vishnupriyan et al. 2021), and (Huang, Wang et al. 2020) datasets).] Please clarify whether this [conference proceeding or publication] was peer-reviewed and formally published. If this work was previously peer-reviewed and published, in the cover letter please provide the reason that this work does not constitute dual publication and should be included in the current manuscript.

3. Please upload a new copy of Figure 7 as the detail is not clear. Please follow the link for more information: https://blogs.plos.org/plos/2019/06/looking-good-tips-for-creating-your-plos-figures-graphics/" https://blogs.plos.org/plos/2019/06/looking-good-tips-for-creating-your-plos-figures-graphics/

Additional Editor Comments:

Although the reviewers have found this study of interest they have raised some concerns about method adopted.

Reviewers' comments:

Reviewer's Responses to Questions

**Comments to the Author**

1. Is the manuscript technically sound, and do the data support the conclusions?

Reviewer #1: Yes

Reviewer #2: Yes

2. Has the statistical analysis been performed appropriately and rigorously? 

Reviewer #1: Yes

Reviewer #2: Yes

3. Have the authors made all data underlying the findings in their manuscript fully available?

Reviewer #1: Yes

Reviewer #2: Yes

4. Is the manuscript presented in an intelligible fashion and written in standard English?

Reviewer #1: Yes

Reviewer #2: Yes

5. Review Comments to the Author

Reviewer #1: Manuscript Number: PONE-D-22-34657

Title: “Mapping CircRNA–miRNA–mRNA Regulatory Axis Identifies hsa_circ_0080942 and hsa_circ_0080135 as a potential theranostic agents for SARS-CoV-2 infection”

This is an interesting small study and the authors have collected a unique dataset using cutting edge methodology. The paper is well written and structured.

The following minor issues should be addressed:

1) In introduction section first paragraph author may update number of total people have been infected and death globally with virus SARS-CoV-2 with updated references.

Reviewer #2: The manuscript entitled Mapping CircRNA–miRNA–mRNA Regulatory Axis Identifies hsa_circ_0080942 and hsa_circ_0080135 as a potential theranostic agents for SARS-CoV-2 infection can be accepted for publication after few improvements

The author have selected the common circRNAs from circRNA datasets, mRNA datasets, and miRNA datasets. How the author found cicRNAs from mRNAs and miRNAs datasets? The authors are suggested to clearly describe the results in detail rather than superficial writing.

Why the author has used differentially expressed mRNA obtained from the mRNA datasets for the GO and KEGG pathway analysis? Why author not preferred the common mRNA which are target of circRNA as well as predicted from mRNA datasets.

The quality of image needs to be improved as picture is blur and not clear

6. PLOS authors have the option to publish the peer review history of their article (what does this mean?). If published, this will include your full peer review and any attached files.

Reviewer #1: **Yes: **Santosh Kumar

Reviewer #2: No

---

## [Author Response · Author response to Decision Letter 0]

23 Feb 2023

Response to reviewers

Dear Editor,

We wish to express our appreciation for your in-depth comments, suggestions and corrections, and we would like to convey our sincere thanks for allowing us to improve our manuscript entitled “Mapping CircRNA–miRNA–mRNA Regulatory Axis Identifies hsa_circ_0080942 and hsa_circ_0080135 as a potential theranostic agents for SARS-CoV-2 infection”. 

Thank you for your very careful review of our paper. A major revision of the paper has been carried out to take all of weaknesses and limitations identified by the respective reviewers into account. And in the process, we truly hope that the revised manuscript is clear enough to follow.

Below is an abridged summary of reviewer’s comments with a detailed response and description of the changes made to the article. Should you find the paper requires further clarification or revision, we most certainly stand ready to do so.

Looking forward to your positive response

Sincerely,

Dr. Faryal Mehwish Awan

Journal Requirements:

Comment # 1

Response

We have formatted our manuscript according to PLOS ONE’s style. The format has been updated in the revised version of the manuscript as per journal requirements. 

Comment # 2

We noted in your submission details that a portion of your manuscript may have been presented or published elsewhere. [All the datasets analyzed during the current study are accessible from the literature as well as from the GEO database (https://www.ncbi.nlm.nih.gov/gds/) with accession details (GSE166552, GSE19137, PRJCA002617, (McDonald, Enguita et al. 2021), (Arora, Singh et al. 2020), (Farr, Rootes et al. 2021), (Li, Hu et al. 2020), (Chow and Salmena 2020), (Demirci and Demirci 2021), (Chi, Ge et al. 2020), (Lin, Luo et al. 2020), (Chen, Wu et al. 2020), (Chen, Liu et al. 2020), (Blanco-Melo, Nilsson-Payant et al. 2020), (Li, Hu et al. 2020), (Del Valle, Kim-Schulze et al. 2020), (Qin, Zhou et al. 2020), (Yang, Shen et al. 2020), (Dhar, Vishnupriyan et al. 2021), and (Huang, Wang et al. 2020) datasets).] Please clarify whether this [conference proceeding or publication] was peer-reviewed and formally published. If this work was previously peer-reviewed and published, in the cover letter please provide the reason that this work does not constitute dual publication and should be included in the current manuscript. 

Response

Yes, we agree with the observation that all the datasets analyzed are already available publically. However, current study does not constitute as dual publication because this study only used the available datasets for novel integrative analysis in order to identify most promising and potential drug targets and to develop appropriate therapeutic strategies. It is important to note that the Gene Expression Omnibus (GEO) repository archives and freely distributes microarray, next-generation sequencing and other forms of high-throughput functional genomics data. Such datasets hold great value for knowledge discovery, particularly when integrated and can potentially bring novel insights into essential questions. The present study has aimed at prioritizing the potential circRNA candidates for a prospective theranostic evaluation via exploring the existing publically available datasets in COVID-19 setting. Public databases have a lot of high throughput data, which greatly helps in revealing the possible disease pathogenesis and identifying potential targets for drug design. Experimental validation of all the discovered associations, let alone all the possible interactions between them, is time-consuming and expensive. In conventional approaches, large experimental screenings are currently used to identify potential leading compounds, but they require significant time and resources. However, one of the lessons learned during the current pandemic is that innovative approaches are required to speed up drug development while increasing its success rate. Since gene expression data are high-dimensional data, an important research aim in the analysis of transcription profiles is the discovery of small subset of biomarkers containing the most discriminant information. Therefore, current study computationally prioritized the data available in the databases for potential SARS-CoV-2 inhibitors by using an integrated approach. 

 A number of recently published research studies have utilized publically available datasets for the prioritization of most promising candidates. Some of these studies are (Dat, et al., 2022; Pandya, et al., 2020; Shams, et al., 2020; Venugopal, et al., 2022; Wu, et al., 2021; Zhang, et al., 2021)

We have added this information in the cover letter to clarify this comment. 

Comment: 3

Please upload a new copy of Figure 7 as the detail is not clear. Please follow the link for more information: https://blogs.plos.org/plos/2019/06/looking-good-tips-for-creating-your-plos-figures-graphics/" https://blogs.plos.org/plos/2019/06/looking-good-tips-for-creating-your-plos-figures-graphics/

Response

We agree with your assessment. We have now followed figure graphics requirements for all the figures and have also uploaded a new copy of Figure 7 as “Fig7.tif”. 

Fig 7: Pathway analysis of COVID-19 pathogenesis (KEGG pathway ID: map05171). Highlighted genes are targets of miRNAs and indirect targets of two prioritized circRNAs 

Comment: 4

Response

We have rechecked the whole manuscript for any errors in the references. Corrections have been made and highlighted. 

Reviewer #1

Comments and Suggestions for Authors

This is an interesting small study and the authors have collected a unique dataset using cutting edge methodology. The paper is well written and structured.

Response

We appreciate the positive feedback from the reviewer and would like to thank the respected reviewer for the encouraging assessment and the comments that helped us to improve the manuscript. 

Comment: 1

The following minor issues should be addressed:

In introduction section first paragraph author may update number of total people have been infected and death globally with virus SARS-CoV-2 with updated references.

Response

We agree with your assessment. As per your suggestion, we have updated total number as per World Health Organization report and revised the statement as follows. 

Currently, >750 million people have been infected globally while >6.8 million people have lost their lives due to COVID-19 (World Health Organization, February 22, 2023).

We have updated this information in the revised version of the manuscript.

Reviewer #2

Comments and Suggestions for Authors

The manuscript entitled Mapping CircRNA–miRNA–mRNA Regulatory Axis Identifies hsa_circ_0080942 and hsa_circ_0080135 as a potential theranostic agents for SARS-CoV-2 infection can be accepted for publication after few improvements

Response

We appreciate the positive feedback from the reviewer and would like to thank the respected reviewer for the encouraging assessment and the comments that helped us to improve the manuscript. 

Comment: 1

The author have selected the common circRNAs from circRNA datasets, mRNA datasets, and miRNA datasets. How the author found cicRNAs from mRNAs and miRNAs datasets? The authors are suggested to clearly describe the results in detail rather than superficial writing.

Response

The target circRNAs of differentially expressed miRNAs were predicted using different comprehensive databases including CircBank, CircInteractome and RNAInter v4.0 web tools (Table 2). CircBank (http://www.circbank.cn/) is a comprehensive, publicly available, functionally annotated human circRNAs database containing information of about 140,000 circRNAs from many different sources (Liu, et al., 2019). The Users can access information regarding conservation status, miRNA targets as well as protein coding potential of query circRNAs (Liu, et al., 2019). CircInteractome (https://circinteractome.nia.nih.gov/) is a readily accessible web tool for mapping miRNAs and protein-binding sites on junctions as well as junction-flanking sequences of human circRNAs (Dudekula, et al., 2016). RNA Interactome Database, RNAInter v4.0 (http://www.rnainter.org/) is a comprehensive RNA-associated interactome platform containing information of more than 41 million interactions of cellular RNAs in 154 species with evidence from both computational and experimental sources (Kang, et al., 2021). Selection criteria, threshold and prediction scores for each database were selected on the basis of their previously reported relationship with low false discovery rate and high accuracy in experimental validation studies via PCR and Luciferase assays. 

For the prediction of potential circRNAs associated with differentially expressed mRNAs, first we predicted miRNAs associated with these mRNA and then circRNAs associated with predicted miRNAs. We used databases including miRDB, miRWalk 2.0, miRTarBase, and TargetScan 7.0 for the prediction of miRNAs associated with respective mRNAs. miRDB (http://mirdb.org/), is an integrative, freely accessible, open platform for the prediction of miRNA targets. miRNA-target interactions with scores ≥80.0 were considered relevant, statistically significant and with higher confidence in the interactions whereas miRNA-target interactions with scores ≤80.0 were considered not relevant. By utilizing high-throughput experimental data, miRDB predicts miRNA targets in five species along with integrative analysis of gene ontology (GO) data (Wang, 2008). miRWalk 2.0 (http://mirwalk.umm.uni-heidelberg.de/) provides information of more than 949 million computationally predicted as well as experimentally validated miRNA-mRNA interactions. In order to ensure reliability and accuracy of forecast results, miRWalk 2.0 incorporates 12 algorithms for prediction including miRWalk, mirbridge, Targetscan, Microt4, PITA, Pictar2, RNAhybrid, RNA22, miRNAMap, miRanda, miRMap and miRDB (Dweep and Gretz, 2015). Cut-off value with a binding score of > 0.95 was considered as a screening threshold. miRTarBase (https://miRTarBase.cuhk.edu.cn/~miRTarBase/miRTarBase_2022/php/index.php) is a manually curated database containing information of more than 360,000 experimentally validated miRNA-mRNA interactions (Huang, et al., 2020). miRNA-mRNA interactions have been validated experimentally using microarray, CLIP-seq technology, reporter assays, high through-put sequencing and western blot experiments (Huang, et al., 2020). All the targets identified via miRTarBase were selected for further analysis. TargetScan v7.0 (http://www.targetscan.org/vert_80/), a flexible web based tool, predicts sequence based effective regulatory targets of miRNAs by incorporating 14 different features (Agarwal, et al., 2015). Conservation aggregate score of > 0.80 was considered as selection criteria as this score provides low false discovery rates. An overlap in at least two databases was used as filtering criteria for prioritizing and considering potential candidate targets. Previous comparative studies conducted on miRNA target prediction programs suggested that no program performed consistently superior to all others. Indeed, it has become a common practice for researchers to look at predictions produced by different miRNA-target prediction programs and focus on their intersection which might enhances the performance of analyses as well as improve prediction precision. The differences between algorithms are mostly seen in their respective weaknesses, i.e., the subset of false positives. For that reason, the fundamental motivation to focus selectively on the shared prediction by two algorithms is to eliminate false positives while preserving the vast majority of true positive RNAs. Therefore, conclusively, predictions are much more reliable when two or more prediction algorithms are combined, and the minimal loss of true positives are greatly outweighed by the removal of false positives. Comprehensive analysis of differentially expressed miRNA datasets revealed 38937 target circRNAs. On the other hand, comprehensive analysis of 5109 predicted miRNAs against differentially expressed mRNAs revealed 858423 circRNAs having binding sites for respective miRNAs.

Comment: 2

Why the author has used differentially expressed mRNA obtained from the mRNA datasets for the GO and KEGG pathway analysis? Why author not preferred the common mRNA which are target of circRNA as well as predicted from mRNA datasets.

Response

The most common complication associated with this COVID-19 is the cytokine storm which is responsible for mortality. Thus, targeting the cytokine storm with new medications is needed to hamper COVID-19 complications. The purpose of performing KEGG pathway analysis was to identify relationship between COVID-19 induced cytokine storm related genes and prioritized circRNAs including hsa_circ_0080942 and hsa_circ_0080135 from final circRNAs-miRNAs-mRNAs axis. The hsa_circ_0080942 network included 47 miRNAs and 15 cytokine storm related genes forming 88 pairs of circRNA-miRNA-mRNA axis. The hsa_circ_0080135 network included 39 miRNAs and 15 cytokine storm related genes forming 77 pairs of circRNA-miRNA-mRNA axis. We proposed that via targeting two prioritized circRNAs, maximum number of cytokines dysregulated during SARS-CoV-2 infection can be targeted as represented from the Figure 7.

Fig 7: Pathway analysis of COVID-19 pathogenesis (KEGG pathway ID: map05171). Highlighted genes are targets of miRNAs and indirect targets of two prioritized circRNAs 

Comment: 3

The quality of image needs to be improved as picture is blur and not clear

Response

We agree with your assessment. We have now uploaded a new copy of Figure 7 as “Fig7.tif”. In addition we have also updated “Table 4” in tabular text form previously uploaded as a figure due to complexity of data. 

Fig 7: Pathway analysis of COVID-19 pathogenesis (KEGG pathway ID: map05171). Highlighted genes are targets of miRNAs and indirect targets of two prioritized circRNAs 

 

Table 4: Datasets used for the analysis of SARS-CoV-2 related cytokines 

Genes Study 1 Study 2 Study 3 Study 4 Study 5 Study 6 Study 7 Study 8 Study 9 Study 10 Study 11 Study 12

 up-regulated down-regulated up-regulated down-regulated up-regulated down-regulated up-regulated down-regulated up-regulated down-regulated up-regulated down-regulated up-regulated down-regulated up-regulated down-regulated up-regulated down-regulated up-regulated down-regulated up-regulated down-regulated up-regulated down-regulated

IL-1β _ _ _ _ _ _ _ _ √ _ _ _ √ _ _ _ _ _ _ _ √ _ _ _

IL-2 √ _ _ _ _ _ _ _ _ _ _ _ √ _ _ _ _ _ _ _ _ _ √ _

IL-4 _ _ _ _ _ _ _ _ _ _ _ _ √ _ _ _ _ _ _ _ _ _ √ _

IL-6 _ _ √ _ √ _ _ _ √ _ √ _ √ _ √ _ √ _ √ _ √ _ √ _

IL-7 √ _ _ _ _ _ _ _ _ _ _ _ _ _ _ _ _ _ √ _ _ _ _ _

IL-8 _ _ _ _ _ _ _ _ √ _ √ _ _ _ √ _ _ _ √ _ _ _ _ _

IL-10 √ _ _ _ √ _ _ _ √ _ _ _ √ _ _ _ _ _ √ _ _ _ √ _

IL-12 _ _ _ _ _ _ _ _ _ _ _ _ _ _ _ _ _ _ _ _ _ _ _ _

IL-13 _ _ _ _ _ _ _ _ _ _ _ _ _ _ _ _ _ _ _ _ √ _ _ _

IL-17 _ _ _ _ _ _ _ _ _ _ _ _ _ _ _ _ _ _ _ _ _ _ _ _

IL-18 _ _ _ _ _ _ _ _ _ _ _ _ _ _ _ _ _ _ √ _ √ _ _ _

IL-23 _ _ _ _ _ _ _ _ _ _ _ _ _ _ _ _ _ _ _ _ _ _ _ _

IL-33 _ _ _ _ _ _ _ _ _ _ _ _ _ _ _ _ _ _ _ _ _ _ _ _

 IL-37 _ _ _ _ _ _ _ _ _ _ _ _ _ _ _ _ _ _ _ _ _ _ _ _

IL-38 _ _ _ _ _ _ _ _ _ _ _ _ _ _ _ _ _ _ _ _ _ _ _ _

TNF-α √ _ _ _ √ _ _ _ √ _ √ _ √ _ √ _ _ _ _ _ _ _ √ _

IFN-γ _ _ _ _ √ _ _ _ _ _ _ _ √ _ _ _ _ _ _ _ _ _ √ _

CCL2 _ _ _ _ _ _ _ _ _ _ _ _ _ _ _ _ √ _ _ _ _ _ _ _

CXCL6 _ _ _ _ _ _ _ _ _ _ _ _ _ _ _ _ _ _ _ _ _ _ _ _

CXCL8 _ _ _ _ _ _ _ _ _ _ _ _ _ _ _ _ √ _ _ _ _ _ _ _

CXCL10 _ _ _ _ _ _ _ _ _ _ _ _ _ _ _ _ √ _ _ _ _ _ _ _

 IP-10 √ _ _ _ _ _ √ _ _ _ _ _ _ _ _ _ _ _ √ _ √ _ _ _

MIP-1A √ _ _ _ _ _ √ _ _ _ _ _ _ _ _ _ _ _ √ _ √ _ _ _

MIP1-B _ _ _ _ _ _ √ _ _ _ _ _ _ _ _ _ _ _ _ _ √ _ _ _

PDGF _ _ _ _ _ _ _ _ _ _ _ _ _ _ _ _ _ _ _ _ √ _ _ _

MCP1 √ _ _ _ _ _ √ _ _ _ _ _ _ _ _ _ _ _ √ _ _ _ _ _

 GM-CSF _ _ _ _ _ _ _ _ _ _ _ _ _ _ _ _ _ _ _ _ _ _ _ _

M-CSF _ _ _ _ _ _ _ _ _ _ _ _ _ _ _ _ _ _ √ _ √ _ _ _

G-CSF √ _ _ _ _ _ _ _ _ _ _ _ _ _ _ _ _ _ √ _ √ _ _ _

FGF _ _ _ _ _ _ _ _ _ _ _ _ _ _ _ _ _ _ _ _ _ _ _ _

HGF _ _ _ _ _ _ _ _ _ _ _ _ _ _ _ _ _ _ _ _ √ _ _ _

TGF-β _ _ _ _ _ _ _ _ _ _ _ _ _ _ _ _ _ _ _ _ _ _ _ _

 (Huang, et al., 2020)

(Chen, et al., 2020)

(Chen, et al., 2020)

(Liu, et al., 2020)

(Qin, et al., 2020)

(Farr, et al., 2021)

(Dhar, et al., 2021)

(Del Valle, et al., 2020)

(Blanco-Melo, et al., 2020)

(Chi, et al., 2020)

(Yang, et al., 2020)

(Lin, et al., 2020)

We hope that our additions to the manuscript will satisfy the reviewers, and thank both reviewers for their precise and insightful comments, and for the careful attention that they have paid which allowed us to improve the manuscript. We look forward to hearing from you regarding our submission. We would be glad to respond to any further questions and comments that you may have.

Reviewer's Responses to Questions

Comments to the Author

1. Is the manuscript technically sound, and do the data support the conclusions?

Reviewer #1: Yes

Reviewer #2: Yes

2. Has the statistical analysis been performed appropriately and rigorously?

Reviewer #1: Yes

Reviewer #2: Yes

3. Have the authors made all data underlying the findings in their manuscript fully available?

Reviewer #1: Yes

Reviewer #2: Yes

4. Is the manuscript presented in an intelligible fashion and written in standard English?

Reviewer #1: Yes

Reviewer #2: Yes

References

Agarwal, V., et al. (2015) Predicting effective microRNA target sites in mammalian mRNAs, elife, 4, e05005.

Blanco-Melo, D., et al. (2020) Imbalanced host response to SARS-CoV-2 drives development of COVID-19, Cell, 181, 1036-1045. e1039.

Chen, G., et al. (2020) Clinical and immunological features of severe and moderate coronavirus disease 2019, The Journal of clinical investigation, 130, 2620-2629.

Chen, L., et al. (2020) Analysis of clinical features of 29 patients with 2019 novel coronavirus pneumonia, Zhonghua jie he he hu xi za zhi= Zhonghua jiehe he huxi zazhi= Chinese journal of tuberculosis and respiratory diseases, 43, E005-E005.

Chi, Y., et al. (2020) Serum cytokine and chemokine profile in relation to the severity of coronavirus disease 2019 in China, The Journal of infectious diseases, 222, 746-754.

Dat, V.H.X., et al. (2022) Identification of potential microRNA groups for the diagnosis of hepatocellular carcinoma (HCC) using microarray datasets and bioinformatics tools, Heliyon, 8, e08987.

Del Valle, D.M., et al. (2020) An inflammatory cytokine signature predicts COVID-19 severity and survival, Nature medicine, 26, 1636-1643.

Dhar, S.K., et al. (2021) IL-6 and IL-10 as predictors of disease severity in COVID-19 patients: results from meta-analysis and regression, Heliyon, 7, e06155.

Dudekula, D.B., et al. (2016) CircInteractome: a web tool for exploring circular RNAs and their interacting proteins and microRNAs, RNA biology, 13, 34-42.

Dweep, H. and Gretz, N. (2015) miRWalk2. 0: a comprehensive atlas of microRNA-target interactions, Nature methods, 12, 697-697.

Farr, R., et al. (2021) Altered microRNA expression in COVID-19 patients enables identification of SARS-CoV-2 infection.

Huang, C., et al. (2020) Clinical features of patients infected with 2019 novel coronavirus in Wuhan, China, The lancet, 395, 497-506.

Huang, H.-Y., et al. (2020) miRTarBase 2020: updates to the experimentally validated microRNA–target interaction database, Nucleic acids research, 48, D148-D154.

Kang, J., et al. (2021) RNAInter v4. 0: RNA interactome repository with redefined confidence scoring system and improved accessibility, Nucleic acids research.

Lin, L., et al. (2020) Long-term infection of SARS-CoV-2 changed the body's immune status, Clinical Immunology, 218, 108524.

Liu, J., et al. (2020) Longitudinal characteristics of lymphocyte responses and cytokine profiles in the peripheral blood of SARS-CoV-2 infected patients, EBioMedicine, 55, 102763.

Liu, M., et al. (2019) Circbank: a comprehensive database for circRNA with standard nomenclature, RNA biology, 16, 899-905.

Pandya, P.H., et al. (2020) Systems biology approach identifies prognostic signatures of poor overall survival and guides the prioritization of novel bet-chk1 combination therapy for osteosarcoma, Cancers, 12, 2426.

Qin, C., et al. (2020) Dysregulation of immune response in patients with coronavirus 2019 (COVID-19) in Wuhan, China, Clinical infectious diseases, 71, 762-768.

Shams, R., et al. (2020) Identification of potential microRNA panels for pancreatic cancer diagnosis using microarray datasets and bioinformatics methods, Scientific Reports, 10, 7559.

Venugopal, P., et al. (2022) Prioritization of microRNA biomarkers for a prospective evaluation in a cohort of myocardial infarction patients based on their mechanistic role using public datasets, Frontiers in Cardiovascular Medicine, 9.

Wang, X. (2008) miRDB: a microRNA target prediction and functional annotation database with a wiki interface, Rna, 14, 1012-1017.

Wu, A.T., et al. (2021) Multiomics identification of potential targets for Alzheimer disease and antrocin as a therapeutic candidate, Pharmaceutics, 13, 1555.

Yang, Y., et al. (2020) Plasma IP-10 and MCP-3 levels are highly associated with disease severity and predict the progression of COVID-19, Journal of Allergy and Clinical Immunology, 146, 119-127. e114.

Zhang, P., et al. (2021) Bioinformatics analysis of candidate genes and pathways related to hepatocellular carcinoma in China: a study based on public databases, Pathology and Oncology Research, 13.

---

## [Decision Letter · Decision Letter 1]

13 Mar 2023

Mapping CircRNA–miRNA–mRNA Regulatory Axis Identifies hsa_circ_0080942 and hsa_circ_0080135 as a potential theranostic agents for SARS-CoV-2 infection

PONE-D-22-34657R1

Dear Dr. Awan,

We’re pleased to inform you that your manuscript has been judged scientifically suitable for publication and will be formally accepted for publication once it meets all outstanding technical requirements.

Kind regards,

Kanhaiya Singh, Ph.D

Academic Editor

PLOS ONE

Additional Editor Comments (optional):

Reviewers' comments:

Reviewer's Responses to Questions

**Comments to the Author**

1. If the authors have adequately addressed your comments raised in a previous round of review and you feel that this manuscript is now acceptable for publication, you may indicate that here to bypass the “Comments to the Author” section, enter your conflict of interest statement in the “Confidential to Editor” section, and submit your "Accept" recommendation.

Reviewer #2: All comments have been addressed

2. Is the manuscript technically sound, and do the data support the conclusions?

Reviewer #2: Yes

3. Has the statistical analysis been performed appropriately and rigorously? 

Reviewer #2: Yes

4. Have the authors made all data underlying the findings in their manuscript fully available?

Reviewer #2: Yes

5. Is the manuscript presented in an intelligible fashion and written in standard English?

Reviewer #2: Yes

6. Review Comments to the Author

Reviewer #2: Following the suggestions, the author has addressed the issues and modified the manuscript accordingly and can be considered for the publication

7. PLOS authors have the option to publish the peer review history of their article (what does this mean?). If published, this will include your full peer review and any attached files.

Reviewer #2: No

---

## [Editor Report · Acceptance letter]

4 Apr 2023

PONE-D-22-34657R1 

Mapping CircRNA–miRNA–mRNA Regulatory Axis Identifies hsa_circ_0080942 and hsa_circ_0080135 as a potential theranostic agents for SARS-CoV-2 infection 

Dear Dr. Awan:

I'm pleased to inform you that your manuscript has been deemed suitable for publication in PLOS ONE. Congratulations! Your manuscript is now with our production department. 

Kind regards, 

on behalf of

Dr. Kanhaiya Singh 

Academic Editor

PLOS ONE